# EvoPress: Towards Optimal Dynamic Model Compression via Evolutionary Search

## Abstract

The high computational costs of large language models (LLMs) have led to a flurry of research on LLM compression, via methods such as quantization, sparsification, or structured pruning. A new frontier in this area is given by *dynamic, non-uniform* compression methods, which adjust the compression levels (e.g., sparsity) per-block or even per-layer in order to minimize accuracy loss, while guaranteeing a global compression threshold. Yet, current methods rely on heuristics for identifying the "importance" of a given layer towards the loss, based on assumptions such as *error monotonicity*, i.e. that the end-to-end model compression error is proportional to the sum of layer-wise errors. In this paper, we revisit this area, and propose a new and general approach for dynamic compression that is provably optimal in a given input range. We begin from the motivating observation that, in general, *error monotonicity does not hold for LLMs*: compressed models with lower sum of per-layer errors can perform *worse* than models with higher error sums. To address this, we propose a new general evolutionary framework for dynamic LLM compression called EvoPress, which has provable convergence, low sample and evaluation complexity. We show that these theoretical guarantees lead to highly competitive practical performance for dynamic compression of Llama, Mistral and Phi models. Via EvoPress, we set new state-of-the-art results across all compression approaches: structural pruning (block/layer dropping), unstructured sparsity, as well as quantization with dynamic bitwidths.

## 1 Introduction

Model compression has become a standard way of reducing the deployment costs of large language models (LLMs). Current post-training compression techniques can be roughly categorized into quantization-based, which reduce the bit-width of weights or activations, e.g. (Frantar et al., 2022; Lin et al., 2023; Dettmers & Zettlemoyer, 2022; Tseng et al., 2024), pruning-based, which sparsify the weight matrices, e.g. (Frantar & Alistarh, 2023; Yin et al., 2024), or structured pruning / layer dropping, which drop entire model components, e.g. (Kim et al., 2024; Men et al., 2024). While constantly improving their performance, existing compression methods are reaching diminishing returns in terms of accuracy-vs-compression (Dettmers et al., 2023; Tseng et al., 2024).

In this context, a new direction is *dynamic*, or *non-uniform*, layer-wise compression, in which different layers can be compressed to various levels, according to their "sensitivity" relative to the model output. Dynamic compression allows to maximize model accuracy while satisfying a given compression requirement, e.g. a target model size. Instance-specific solutions for this problem have already been proposed for essentially every compression type: sparsity (Yin et al., 2024), quantization (Frantar & Alistarh, 2022), or layer dropping (Kim et al., 2024; Men et al., 2024). Broadly, these approaches work by assigning an *error/sensitivity score* function to each layer and compression level, which measures the impact of its compression on output loss increase. Then, one calculates a compression assignment which minimizes the sum of error scores, while still satisfying the global compression constraint. Thus, such approaches inherently assume *error monotonicity*: i.e., that a lower sum of error scores implies a lower compression error for the entire model.

Our work starts from the observation that error monotonicity *does not hold* generally for LLM compression: specifically, there are instances where *compressed models with lower sums of per-layer errors can perform* worse *than models with higher error*. We illustrate this fact in Table 1,

Table 1: Depth pruning is not monotone. In this example (Llama-3-8B with Fineweb-Edu calibration), removing strictly more blocks (depicted in orange) *can improve* perplexity across sources. Left half of block corresponds to attention layer, right half to MLP.

| Model | Configuration (Each block contains Attention + MLP) | Wiki2↓ | C4↓ | FW↓ |
|-------|------------------------------------------------------|--------|-----|-----|
| Llama-3-8B | | 5.54 | 8.80 | 7.72 |
| | | 188.01 | 147.25 | 70.46 |
| | | **24.39** | **35.53** | **26.24** |

which shows an instance of a layer dropping configuration where keeping *more blocks* leads to massively higher perplexity than an instance which prunes *strictly less* blocks.

**Contribution.** This refutation of error monotonicity implies that most prior approaches, which are based on this assumption, can lead to sub-optimal solutions. Thus, it motivates our investigation of alternatives towards optimal non-uniform compression. For this, we propose a new evolutionary search approach called EvoPress, which is provably convergent, and is also sample and iteration efficient. Thus, EvoPress is the first non-uniform compression method with guarantees; its two efficiency properties are critical for practicality in the context of LLMs, where the cost of evaluating single models ("offspring") is exceedingly high. We validate the approach across all three popular approaches for post-training LLM compression: layer dropping, one-shot sparsification, and quantization. We find that EvoPress consistently improves upon existing techniques, with major improvements at higher compression ratios.

In more detail, we assume a setting where we are given a pre-trained model, a compression constraint such as the target model size, a set of compression options (e.g., 10 possible sparsity options per layer), and aim to identify a per-layer assignment which satisfies the constraint, while minimizing accuracy loss, measured in perplexity or in-context learning accuracy degradation. As is standard, e.g. (Frantar & Alistarh, 2022), from the compression options we build a *level database*, where each layer is compressed independently to each compression option. During the candidate search, our *offspring* are models stitched together from the level database, and our *fitness function* will be the difference (e.g., in KL-divergence) between the outputs of the offspring and the original model, on a set of calibration samples.

At each step, our search algorithm starts with a single search point (candidate model), and generates a constant $\lambda \geq 1$ additional offspring, by applying a mutation operation which preserves the compression constraint. The selection stage is composed of multiple steps, where we *iteratively* evaluate the offspring and parent on *increasingly many* randomly chosen samples. For instance, we may start to evaluate the parent and $\lambda = 64$ offspring *on less than a single sample* on the first sub-step, but progressively multiply the number of calibration samples as we sift through candidates, reducing variance as we obtain more competitive offspring. We found this trade-off between exploration and evaluation variance essential for efficiency on LLMs, as it drastically reduces our total number of evaluations relative to the case where all the initial offspring must be evaluated on a full batch.

Our algorithm guarantees convergence: specifically, any linear fitness function[1] defined on the $n$-dimensional hypercube will be maximized in expected $O(k(n-k)/\lambda)$ generations under the constraint $|x|_1 = k$, where $\lambda$ is the number of offspring. The proof is quite non-trivial, as it needs to adapt stochastic drift analysis techniques, via a novel potential function, to the case where multiple offspring are examined in each sub-step. In Figure 1, we illustrate the algorithm's fast convergence and high efficiency on a practical example with correlated block dropping on Llama-3-8B, where we determined the optimum via (expensive) exhaustive search: EvoPress is able to reach the optimum in only 6 generations, using a total of only 56 model evaluations.

A key advantage of our approach is that it is agnostic of the model architecture and compression type. We illustrate this in our experimental results, which are the first to span all three compression methods, across different LLM families. Specifically, results show that EvoPress significantly improves upon all prior work on depth pruning in terms of accuracy-vs-compression, especially

---

[1]The class of linear functions is a classical benchmark for randomized search heuristics and theory of evolutionary algorithms, e.g. (Droste et al., 2002), (Doerr & Künnemann, 2015), (Lengler & Spooner, 2015).

at medium levels, and also outperforms the prior best methods–OWL and dynamic programming, respectively–for non-uniform pruning and quantization. Moreover, it can do so efficiently: the full version of EvoPress converges in a few hours on a single RTX 3090 GPU, and we also present a lightweight version which utilizes fewer samples and converges in $\sim 1$ hour in the same setting.

## 2 RELATED WORK

To our knowledge, we are the first to present a unified approach which covers all types of post-training LLM compression (i.e., layer dropping / depth pruning and non-uniform pruning / quantization)–so far, these problems have generally been approached independently.

**Depth Pruning.** Recently, there has been a lot of interest in compression by removing entire Transformer blocks, both for efficiency and to gain insights about the language model itself. Most methods are based on scoring the importance of each block, and then maximizing the importance of the resulting model by removing the blocks of lowest importance. Weight Subcloning (Samragh et al., 2023) proposed a multi-step process to find good initializations for an untrained smaller model given an already trained larger one, where the importance of each block is scored based on the ratio of $\ell_2$ norms between the output embeddings of the block with and without the residual connection. Shortened Llama (Kim et al., 2024) proposes scoring each block by measuring the perplexity after removing the respective block from the full model. ShortGPT (Men et al., 2024) uses the cosine similarity between the input and output embeddings of each block to assess its importance. By contrast, Gromov et al. (2024) restrict themselves to removing *consecutive blocks*, and score each of these removal configurations using cosine similarity.

**Non-Uniform Pruning and Quantization.** He et al. (2018); Ashok et al. (2018) were among the first to consider automatic optimization of non-uniform compression, specifically for the case of pruning, where developed Reinforcement Learning (RL)-based approaches. However, both approaches suffer from high tuning complexity and would be very hard to scale to large models. Follow-up work (Hubara et al., 2021; Yao et al., 2021; Li et al., 2021) considered a similar problem specifically for quantization, but explore computationally-expensive solvers (e.g. ILPs) which rely on the fact that quantization has only a small number of choices (precision levels) per layer. SPDY (Frantar & Alistarh, 2022) considered a unified framework which reduces the problems to knapsack-type instances, and solves them optimally modulo discretization. However, SPDY explicitly relies on monotonicity and linearity assumptions on the dependency between the per-layer errors and model output error, which we find not to hold on large models, especially in the high-compression regime (e.g., below 3 bits per parameter). Relative to SPDY, EvoPress provides guarantees for a much broader class of input functions, and focuses on efficiency for LLM compression.

The recent OWL method (Yin et al., 2024) focuses on non-uniform pruning of LLMs, and provides consistent improvements over uniform profiles via a layer scoring system which analyzes the activation outlier structure, but does not have any theoretical guarantees. Experimentally, we find that OWL is effective especially for Llama-family models (Touvron et al., 2023) and at moderate sparsities, but observe significant gaps in favor of EvoPress across all models and compression levels.

**NAS and Structural Pruning.** Random search is also popular in the context of structural pruning and Neural Architecture Search (NAS) (Chen et al., 2020; Dong et al., 2021; Wang et al., 2020; Xu et al., 2021; Yin et al., 2021; Molchanov et al., 2022; Kurtić et al., 2024). However, such methods also rely heavily on re-training and have notoriously high costs, which limits their applicability to post-training compression of LLMs. Due to its low sample complexity, we believe that EvoPress could be extensible to lightweight NAS as well, and plan to investigate this in future work.

## 3 METHOD

All applications of EvoPress are grounded in a unified framework, where the objective is to identify the optimal model that adheres to a specified compression method and constraint. Formally, given a base model $M$, we seek to maximize the performance of the compressed model while satisfying the compression constraint:

$$\hat{M}^* = \arg\max_{\hat{M}} f(\hat{M}_v) \quad \text{subject to} \quad g(\hat{M}) \leq C,$$

where $f(\hat{M})$ quantifies the performance of the compressed model $\hat{M}$ and $g(\hat{M})$ represents the compression constraint. For simplicity, we will define $g$ as the model's total size (in terms of parameters); however, the proposed method can be readily adapted to accommodate other practical constraints, such as inference speed.

We approach this optimization problem using evolutionary search, which is a specific form of randomized search. The feasibility of such an approach heavily depends on two factors: the time required to evaluate the fitness of a candidate solution and the number of such function evaluations needed until a satisfying result is achieved. This poses a particular challenge in our case, as assessing the performance of an LLM involves substantial computational costs.

**Level Database.** As a first step, we compress the model to different levels. It is crucial that the units we search over – specifically layers or blocks – are compressed independently; otherwise, we risk losing performance when stitching together the compressed model. Ideally, the difference between two compression levels should be consistent across layers. This uniformity simplifies the optimization process, allowing for the free exchange of compression levels, as we will demonstrate for unstructured sparsity. However, this restriction is not essential for the search procedure to effective. In the context of quantization we will demonstrate a relaxation of this requirement, where compression steps are uniform only across layers of same size.

**Fitness Environment.** Given the specified database, any compressed model is completely characterized by its compression level for each unit (per layer or per block). With $n$ units, each available in $m$ compression levels, our objective is to find

$$\hat{M}^* = \arg\max_{v \in [m]^n} f(\hat{M}_v) \quad \text{subject to} \quad g(\hat{M}_v) \leq C,$$

where we are searching over the set of $n$-tuples over $[m]$. Assessing the performance of a model in practice typically involves benchmark tasks, which have limited scope and require lengthy evaluation. We address these challenges by using the base model as the gold standard and focusing solely on the relative degradation of our compressed models. To quantify this degradation, we measure the Kullback-Leibler (KL) divergence between the two models, as it has proven particularly robust with limited data. Empirically, we observed that already around 65536 tokens of calibration data (corresponding to 8 full sample sequences for Llama-3-8B) are sufficient to reliably determine the quality of the lightweight model. To avoid confusion, we will refrain from inverting the fitness function and from now on consider the minimization problem

$$\hat{M}^* = \arg\min_{v \in [m]^n} D_{KL}(P_M \parallel Q_{\hat{M}_v}) \quad \text{subject to} \quad g(\hat{M}_v) \leq C,$$

where we speak of *higher fitness* whenever the KL-Divergence is *lower*.

**Algorithm.** EvoPress starts from upon the classic $(1+\lambda)$-evolutionary framework, which maintains a single search point at any given time. In each generation, $\lambda$ offspring are generated by copying the parent and then applying a mutation operator to each copy. The offspring are then evaluated on the fitness function, and the fittest one is selected. As an *elitist* evolutionary algorithm, the $(1 + \lambda)$-EA replaces its parent only if the best offspring has superior fitness.

We change this standard algorithm in two important ways. The first is by introducing *level-switch mutation*, a simple mutation operator that ensures high locality while preserving the compression constraint. The operator involves first randomly selecting one unit and increasing its compression level. Next, a second unit is sampled until one with a matching level step size is found, and its compression level is decreased. This approach ensures that 1) the compression constraint is preserved, and 2) the offspring model maintains high similarity to the parent model – an important feature for achieving rapid convergence.

The second modification is that we employ a very aggressive form of *multi-step selection*. In the first stage, all $\lambda$ offspring are evaluated using only a fraction of a full sample. From this, only a small subset of the fittest offspring are selected to compete in the next stage, where they are evaluated on a significantly larger sample size. This process is repeated once more, and in the final stage, the few remaining offspring are evaluated against the parent using a "full" minibatch, consisting of approximately 20-50 times the number of tokens used in the first stage.

---

**Algorithm 1:** EvoPress: A $(1 + \lambda)$-Evolutionary Algorithm with Level-Switch Mutation and Multi-Step Selection for Maximizing a Fitness Function $f : [m]^n \to \mathbb{R}$

---

**Initialization:** $candidates \leftarrow []$ ;
**for** $i \leftarrow 1$ **to** $C$ **do**
    // Only for non-integer target compression
    $candidate \leftarrow sampleUniformly()$;
    $candidates.append(candidate)$;
$x^{(1)} \leftarrow selectTopKFittest(candidates, initialTokens, K = 1)$;
**Optimization:** **for** $t \leftarrow 1$ **to** $\infty$ **do**
    $offspring \leftarrow []$;
    **Mutation:** **for** $i \leftarrow 1$ **to** $\lambda$ **do**
        $y_i \leftarrow x^{(t)}$;
        $y_i \leftarrow LevelSwitchMutation(y_i)$;
        $offspring.append(y_i)$;
    **Selection:** **for** $step \leftarrow 1$ **to** $selectionSteps$ **do**
        **Elitism:** **if** $step = selectionSteps$ **then**
            $offspring.append(x^{(t)})$;
        $offspring \leftarrow selectTopKFittest(offspring, tokens[step], K = survivors[step])$;
    $x^{(t+1)} \leftarrow offspring[0]$;

---

For initialization, we apply the target level directly if it matches an available setting (e.g., all layers at 70% sparsity for an average of 70% sparsity). If the target falls between two compression levels (e.g., for block dropping), we initialize by randomly sampling candidates with some units compressed to the next lower level, and others to the next higher level, selecting the fittest among them. A summary of this optimization procedure can be found in Algorithm 1.

**Design Considerations.** Randomized search heuristics are heavily influenced by the exploration-exploitation dilemma, i.e. trade-off between exploring a broader solution space and intensifying the search around the currently-best solutions. In evolutionary search, many applications utilize sophisticated techniques, such as genetic algorithms, to enhance exploration, which often maintain a large population, introduce crossover operations, and adopt non-elitist strategies, where parents have no chance of survival into the next generation. However, implementing these approaches for LLM compression would come with significant computational costs.

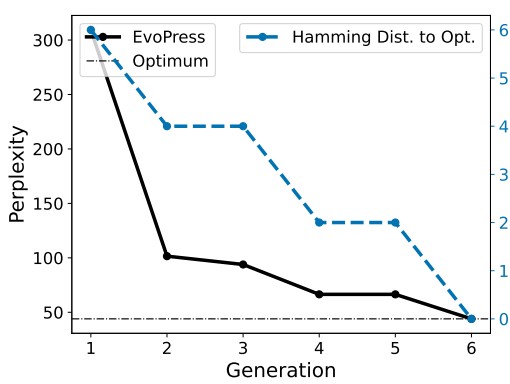

Figure 1: Removing twelve transformer blocks from Llama-3-8B under the constraint that only pairs of consecutive blocks can be removed. Evo-Press finds the optimal configuration from the 8008 possible removal combinations in generation 6.

Crossover, for instance, is only effective if population diversity is preserved, often measured by the sum of pairwise Hamming distances between individuals (Jansen & Wegener, 2002; Opris et al., 2024). While this promotes more thorough exploration of the search space, it requires allocating resources to less promising regions, which may slow progress toward optimal solutions. Similarly, non-elitist algorithms, despite their ability to escape local optima (Dang et al., 2021; Jorritsma et al., 2023; Lengler et al., 2024), also incur costs by frequently discarding potentially useful individuals.

**Convergence.** Contrary to many real-world problems, dynamic model compression with a carefully designed level database creates a notably smooth fitness environment. This is because small changes in the compressed model tend to lead to small changes in performance. Although the search

space expands exponentially with the number of units we search over, the maximum Hamming distance increases only linearly. Therefore, as long as we receive a "signal" indicating the direction of improvement, even with seemingly limited progress per generation, we can converge rapidly to a high-quality solution.

To illustrate this, we consider the problem of removing pairs of consecutive blocks of Llama-3-8B. We perform a brute-force search over all possible 8008 block removal configurations, where six pairs of blocks are removed. Our method identifies the optimal configuration by the 6th generation, having evaluated only 16 candidates for initialization and 8 candidates per generation, using significantly fewer tokens. Figure 1 illustrates how the algorithm progressively approaches the optimum in terms of Hamming distance.

Consequently, our method is heavily exploitation-focused: we rely on elitism, introduce minimal mutation, maintain only a single offspring and therefore employ zero population diversity. We present ablations and a short discussion on these choices in Appendix B.1. EvoPress excels at optimizing smooth fitness environments, a capability we theoretically support by proving rapid convergence under an $\ell_1$-constraint for the class of linear functions.

**Theorem 1.** *Let $n, k \in \mathbb{N}$ with $k \leq n$ and consider the $(1 + \lambda)$-EA with $\lambda \in O(n/\log(n))$ and level-switch mutation. Then any linear fitness function $f : \{x \mid x \in \{0,1\}^n, |x|_1 = n - k\} \to \mathbb{R}$ is optimized in expected*

$$O\left(k \cdot (n - k) \cdot \frac{1}{\lambda}\right) \text{ generations.}$$

**Discussion.** The proof is quite non-trivial, as it builds upon stochastic drift analysis; it is presented in Appendix A. The derived bound has several practical implications. By increasing the number of offspring per generation, we can reduce the number of generations required for convergence, with the reduction scaling proportionally to $\lambda$ up to a reasonably large value. Since our approach uses a highly aggressive form of multi-step selection, the benefit is not simply a zero-sum trade-off. Evaluating many offspring in each generation incurs a significantly lower per-offspring computational cost, leading to a substantial speedup in convergence time. This makes the algorithm highly effective in smooth fitness environments, making it particularly well-suited for dynamic model compression.

## 4 EXPERIMENTS

We now validate the efficiency of EvoPress for determining the optimal layer-wise compression across three approaches: (1) **layer dropping**, where the goal is to isolate the "optimal" set of blocks to drop given a target ratio, (2) **non-uniform unstructured sparsity** and (3) **non-uniform quantization**, where we are given a set of compression options per layer (sparsities or bit-widths), and the goal is to find the "optimal" configuration that matches a certain model size. We focus on LLM compression, given the major interest in reduction of their model size and inference latency, but our method is general and can be applied to any neural network architecture and application domain.

**Experimental Setup.** We consider base models from the Llama-2 and Llama-3 (Touvron et al., 2023) families, Mistral-v0.3 (Jiang et al., 2023), and the instruction-tuned Phi3-Medium-instruct-128k model (Abdin et al., 2024), and adopt KL-divergence as our fitness function as it provides a stronger and more robust signal, reflecting the predictive distribution of the original model. We present ablations to validate this choice in Appendix B.3.

Concretely, our algorithm works as follows: given a uniform or random initial configuration, for each step, we generate new offspring by making random flips, sampled from `min(randint(1,3), randint(1,3))` (increase / decrease) of compression levels under the constraint of fixed overall compression ratio. Initially, we produce a large number of configurations (64-128 in most experiments) and evaluate each on a few data samples (a single sequence on the first round). We choose the top-$k$ best configurations and run the next selection round with fewer candidates and more samples. Finally, we take the best configuration (including the parent) and adopt the best found configuration for the next round. We run for a fixed number of iterations, chosen so that performance on held-out data no longer improves.

To perform per-layer compression via unstructured sparsity and quantization we adopt the data-aware compression methods SparseGPT (Frantar & Alistarh, 2023) and GPTQ (Frantar et al., 2022),

respectively, requiring a calibration set. For this purpose, we utilize Fineweb-Edu (Penedo et al., 2024) as a source of clean and diverse calibration data. Following Egiazarian et al. (2024), we fix the total number of calibration tokens to 8 million (8M). For a fair comparison, all competitive methods employ the same calibration data. The code is attached as supplementary material.

**Evaluation.** We adopt standard LLM evaluation protocol from Frantar et al. (2022). Specifically, we measure the Perplexity metric on the WikiText-2 (Merity et al., 2016) and C4 (Raffel et al., 2019) for language performance and Accuracy on zero-shot evaluations on standard benchmarks: WinoGrande (Sakaguchi et al., 2021), PiQA (Tata & Patel, 2003), HellaSwag (Zellers et al., 2019), ARC-easy and ARC-challenge (Clark et al., 2018) via the LM Eval Harness (Gao et al., 2021).

### 4.1 APPLICATION 1: DEPTH PRUNING

As a first application, we apply EvoPress on Depth Pruning. Although removing entire transformer blocks generally results in greater performance losses compared to other compression techniques, this approach recently attracted attention in the context of initializing smaller models, as it guarantees speedups proportional to the sparsity (Samragh et al., 2023; Kim et al., 2024). Additionally, block dropping provides insights into the capabilities of transformer models, making it relevant for interpretability. We will compare against the following baselines:

- **Shortened Llama** (Kim et al., 2024): Scores blocks on the perplexity change after removal.
- **ShortGPT** (Men et al., 2024): Blocks are scored based on the average cosine similarity between input and output embeddings, including the residual stream.
- **Weight Subcloning** (Samragh et al., 2023): Blocks are scored using the ratio $||f(x)||/||f(x) + x||$, where $x$ is the input embedding and $f(x)$ is the block's output, excluding the residual stream.
- **Sliding Window Cosine Similarity** (Gromov et al., 2024): Sets of consecutive blocks are scored based on the cosine similarity between embeddings before and after the blocks, including residual stream.

While Gromov et al. (2024) directly scores entire removal configurations, Shortened Llama, Short-GPT, and Weight Subcloning determine block removals based on their isolated scores.

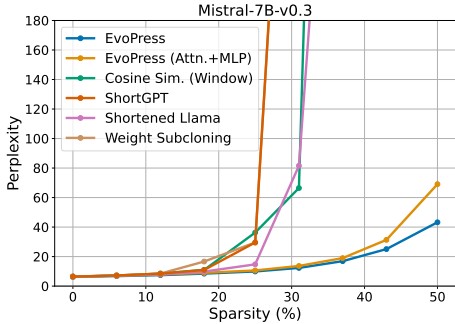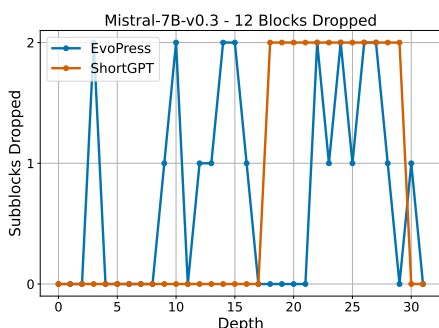

Figure 2: Depth pruning results, on Mistral-7B-v0.3. (Left) Relative to all prior methods, EvoPress shows significantly lower PPL gap relative to the uncompressed model, with remarkably large gaps at medium compression rates. (Right) Examining the blocks dropped, we observe that EvoPress isolates completely different profiles relative to ShortGPT (which scores by cosine similarity).

**Search space.** In our approach, attention and MLP modules are treated independently rather than as a single unit. For each module, there are two options: either retain it or remove it. To achieve a target sparsity/depth, we initially remove an equal number of attention and MLP modules. During mutation, we allow compression level adjustments only between modules of the same type. We leave it open for future research to remove this constraint to allow flexibility in the number of removed attention and MLP modules.

**Experimental results.** Figure 2 compares our method with baselines from previous work on Mistral-7B-v0.3. EvoPress consistently outperforms all previous methods, showing significant im-

provements even at medium sparsity levels. While all baseline methods fail entirely beyond 31.25% sparsity, EvoPress identifies functional submodels even when removing half of the model. To our knowledge, this is the first method to achieve such results. We observed similar collapses in Llama-2-7B and Llama-3-8B, although at slightly higher sparsity. Overall, EvoPress consistently outperforms all baselines across all tested models and sparsities (see Appendix D.1 for full results).

All four previous methods rely on human-crafted scoring methods to identify the optimal combination of transformer blocks to remove. However, these approaches are not only suboptimal, but also prone to bias, as their results may reflect the characteristics of the method itself rather than the model's true behavior. Specifically, we found that most scoring methods tend to favor deeper blocks, resulting in highly similar removal configurations across different prior scoring methods (Appendix 12). This likely occurs because methods that bias towards deeper blocks generally perform better than those that focus on earlier blocks, although neither may be optimal. In contrast, EvoPress employs an unbiased approach, offering more accurate and meaningful insights into the model. As shown in Figure 2, we found that the deeper layers are not necessarily the least important, contradicting conclusions drawn in prior work (Gromov et al., 2024; Men et al., 2024).

## 4.2 APPLICATION 2: UNSTRUCTURED SPARSITY

Next, we examine performance for *unstructured sparsity*, which offers more fine-grained compression. The standard approach is to allocate sparsity *uniformly across layers*. However, some layers may be more sensitive to sparsity, which can significantly impact the model's output. To address this, OWL (Yin et al., 2024) introduces the Layer Outlier Distribution (LOD) metric as a measure of layer saliency, and computes a sparsity profile that is weighted by LOD. A third approach that is vary similar to SPDY (Frantar & Alistarh, 2022), which we also implement as a baseline, is to minimize Normalized Mean Squared Error (NMSE), defined as $\text{NMSE} = \|\hat{Y} - Y\|_2^2 / \|Y\|_2^2$, where $Y$ representing the original model output at a layer, and $\hat{Y}$ the output of the compressed model. Then, the optimal sparsity profile for a given total sparsity can then be determined via a dynamic programming (DP) approach. (The full SPDY method applies a second iterative random search step, which is very expensive to implement at LLM scale, and is therefore omitted.) We compare EvoPress with uniform, OWL, and the DP approach in SPDY. For OWL we used the same hyperparameter grid as the original work and took the configuration yielding best perplexity for each model.

**Search space.** Sparsity levels are generated as follows: For each layer, we first produce the base level corresponding to the targeted average sparsity. Then, we generate both higher and lower compression levels, where the difference between two levels corresponds to a fixed number of weights. In our experiments, we used a "step size" of 1M weights uniformly. This approach enables the mutation of compression levels across all layers, independently of their size. We adopt SparseGPT (Frantar & Alistarh, 2023) as a fast and efficient one-shot layer pruner.

Table 2: Performance of various methods at 70% average sparsity. EvoPress outperforms prior methods both in terms of validation perplexity (PPL) and zero-shot accuracy.

| Model | Sparsity | Wiki2↓ | C4↓ | ArcC↑ | ArcE↑ | HS↑ | PiQA↑ | WG↑ | Avg↑ |
|---|---|---|---|---|---|---|---|---|---|
| | Dense | 5.12 | 6.93 | 43.4 | 76.3 | 57.1 | 78.1 | 69.0 | 64.8 |
| Llama-2-7B | Uniform | 46.51 | 45.30 | 23.1 | 48.4 | 32.4 | 61.3 | 57.1 | 44.5 |
| | DP | 162.12 | 127.88 | 21.5 | 35.0 | 28.6 | 55.7 | 50.3 | 38.2 |
| | OWL | 18.98 | 19.55 | 28.0 | 55.1 | 39.0 | 66.5 | 63.6 | 50.4 |
| | EvoPress | **15.32** | **15.70** | 29.5 | 59.8 | 41.5 | 68.4 | 62.8 | **52.4** |
| | Dense | 5.54 | 7.10 | 50.4 | 80.1 | 60.2 | 79.7 | 72.6 | 68.6 |
| Llama-3-8B | Uniform | 85.84 | 98.35 | 22.7 | 49.9 | 31.4 | 62.1 | 54.4 | 44.1 |
| | DP | 116.91 | 149.13 | 22.6 | 45.9 | 31.3 | 60.6 | 52.5 | 42.6 |
| | OWL | 48.07 | 52.32 | 27.0 | 54.9 | 36.6 | 65.1 | 58.6 | 48.4 |
| | EvoPress | **28.76** | **33.72** | 28.9 | 56.7 | 38.6 | 68.0 | 61.7 | **50.8** |
| | Dense | 4.02 | 8.31 | 60.9 | 84.1 | 64.0 | 81.0 | 76.2 | 73.2 |
| Phi-3-Medium-14B | Uniform | 16.66 | 24.73 | 36.9 | 70.6 | 40.0 | 69.4 | 65.8 | 56.5 |
| | DP | 36.03 | 60.54 | 27.1 | 59.4 | 35.2 | 65.1 | 58.7 | 49.1 |
| | OWL | 15.66 | 23.38 | 35.7 | 69.2 | 39.4 | 68.3 | 64.4 | 55.4 |
| | EvoPress | **13.83** | **19.13** | 41.5 | 73.0 | 43.6 | 71.8 | 69.1 | **59.8** |

**Experimental results.** We compare different methods for non-uniform pruning for 50%, 60% and 70% unstructured sparsity. We report the 70% results in Table 2; the 50% and 60% results can be found in Appendix Tables 13 and 14, respectively. As illustrated in Table 2, EvoPress successfully finds better profiles than uniform sparsity ($3\times$ Perplexity reduction) and noticeably outperforms all other competitive methods (OWL and DP) on PPL and zero-shot average accuracy, by large margins, especially on the larger Phi3 model. The DP solution performs worse than uniform, suggesting that either normalized per-layer error is not a good saliency metric, or that the additive error metric is invalid in this case as well (or both). Examining sparsity profiles (Appendix Figures 9 and 10), we observe that EvoPress prunes the first blocks less aggressively, blocks in the second half of the model more aggressively while keeping the last block relatively dense. Further, EvoPress exhibits stronger deviations from uniform relative to DP and OWL, suggesting it performs broader exploration.

**Running Time.** EvoPress is also time-efficient. Figure 3 illustrates the rapid convergence of our method vs. iterations and time, with smooth and steady improvements in test perplexity. Moreover, we found that, by significantly reducing the number of tokens used in the multi-step selection evaluation, by $4\times$ in the first step and $8\times$ in the last step, and making each generation have fewer offspring, we can significantly speed up the search. This "super-fast" version converges in a little over one GPU hour to similar test PPL (Figure 3, right), demonstrating the sample-robustness of EvoPress, which can lead to further efficiency gains.

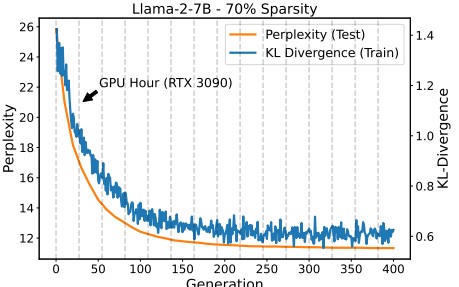 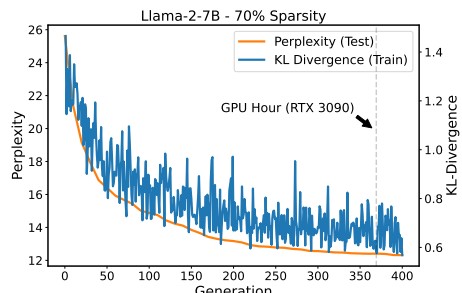

Figure 3: **Left**: The convergence of EvoPress vs. number of generations and wall-clock time (on a single RTX 3090 GPU with 24GB RAM) for Llama-2-7B. We observe convergence close to optimum in 5-6h; **Right**: Convergence of the "superfast" version which reduces the number of tokens used for each evaluation. It converges to similar accuracy in little over one hour, in the same setting.

### 4.3 APPLICATION 3: QUANTIZATION

Finally, we apply evolutionary search to the more challenging problem of non-uniform neural network quantization, where *uniform per-layer quantization* is the most widely-adopted baseline (Frantar et al., 2022; Lin et al., 2023; Chee et al., 2023). However, one could expect that different layers exhibit different sensitivity to quantization, as for unstructured sparsity. As baselines, we consider uniform and DP search defined above. (While OWL has also been applied to quantization, the authors found that it underperforms even relative to uniform per-layer quantization (Yin et al., 2024).) We create configurations with varying bitwidths and run EvoPress to determine the optimal configuration for target compression ratio.

**Search space.** For each linear layer, we produce different configurations via GPTQ (Frantar et al., 2022) with a standard group size of $128$. On each step of evolutionary search, one increases bitwidth in some layers chosen at random while decreasing it in others. To facilitate uniform transitions between compression levels, quantization options differ by integral bits (1 bit in the following). Since different layers may have different sizes, we allow sweeps only between the projections with the same number of elements (i.e. only between MLP and Attention projections).

**Experimental results.** Below, to validate the efficiency of evolutionary search, we consider the challenging problem of quantization to 3 bits and below. For this compression rate, uniform GPTQ quantization faces significant performance drops, motivating more elaborate quantization bitwidth allocation. We produce configurations with 2,3,4,5, and 6 bits and search for an optimal compression profile with respect to the fitness function. Results in Table 3 suggest that non-uniform quantization yields superior quality to baseline options. We visualize quantized configuration found by EvoPress

Table 3: Performance of various profiles at 3-bit quantization, for PPL and avg. zero-shot accuracy.

| Model | Sparsity | Wiki2↓ | C4↓ | ArcC↑ | ArcE↑ | HS↑ | PiQA↑ | WG↑ | Avg↑ |
|---|---|---|---|---|---|---|---|---|---|
| Llama-2-7B | Dense | 5.12 | 6.93 | 43.4 | 76.3 | 57.1 | 78.1 | 69.0 | 64.8 |
| | Uniform | 6.16 | 7.96 | 39.5 | 73.9 | 54.1 | 76.5 | 66.5 | 62.1 |
| | DP | 6.70 | 8.31 | 38.9 | 72.4 | 53.5 | 76.4 | 65.9 | 61.4 |
| | EvoPress | **5.70** | **7.87** | 40.4 | 75.0 | 54.7 | 77.1 | 68.1 | **63.1** |
| Llama-3-8B | Dense | 5.54 | 7.10 | 50.4 | 80.1 | 60.2 | 79.7 | 72.6 | 68.6 |
| | Uniform | 12.19 | 15.76 | 35.2 | 66.9 | 54.0 | 75.2 | 69.6 | 60.2 |
| | DP | 29.00 | 20.03 | 39.8 | 72.0 | 52.9 | 74.7 | 67.2 | 61.3 |
| | EvoPress | **7.49** | **12.03** | 43.0 | 76.4 | 55.4 | 77.3 | 69.7 | **64.3** |
| Phi-3-Medium-14B | Dense | 4.02 | 8.31 | 60.9 | 84.1 | 64.0 | 81.0 | 76.2 | 73.2 |
| | Uniform | 5.18 | 9.05 | 55.1 | 81.6 | 60.8 | 78.9 | 73.6 | 70.0 |
| | DP | 5.72 | 9.71 | 54.7 | 80.4 | 58.4 | 78.6 | 73.5 | 69.1 |
| | EvoPress | **5.09** | **9.00** | 56.7 | 82.6 | 61.0 | 79.2 | 74.7 | **70.8** |

for Llama-3-8B in Appendix Figures 12 and 13. Specifically, we observe that the last block is compressed less aggressively and EvoPress treats `v_proj` as more important than `k_proj` [2].

Overall, we observe that, in this case as well, EvoPress yields significant accuracy improvements (e.g., 1 and 4.1 points on the zero-shot averages on Llama-2 and Llama-3, respectively), compared to the uniform profile. Moreover, the improvement over the next-best method is always significant, both in terms of PPL and zero-shot accuracy.

## 5 CONCLUSION

We have presented EvoPress, a unified optimization framework for non-uniform compression. Evo-Press is based on a new provably-convergent evolutionary search algorithm with low sample and iteration complexity, that is especially well-suited to the loss landscapes arising in LLM compression. Specifically, we have shown that EvoPress can converge extremely fast to accurate configurations for various non-uniform LLM compression problems, and is also fast to execute in practice. We also emphasize the breadth of our study, our method was implemented and tested on three different compression approaches, relative to prior work which largely focused on a single application. Experimental results showed that EvoPress consistently outperforms prior dynamic compression approaches, across all compression types, with large gaps at medium to high compression.

**Limitations.** One interesting direction we did not investigate is the possibility of combining *different compression approaches* into the same search space. This would require changes to our switch mutation strategy, but should be feasible in most cases. Second, we did not investigate finer-grained structured pruning (i.e., removing rows and columns from the weight matrices), as it usually requires extensive retraining to recover accuracy. We plan to investigate this in future work, as our approach is well-suited to it. Finally, we plan to extend our compression results for quantization, to show end-to-end speedups in the context of an inference engine supporting multiple compressed formats, such as vLLM (Kwon et al., 2023).

**Impact Statement.** We presented work that aims to advance efficiency in machine learning. We believe that model compression optimization is a step toward democratizing large-scale model inference, and thus provides opportunities to foster both the development of new applications and the research in the field. There are several important societal concerns about the rapidly growing use of artificial intelligence, but we feel that none of them specifically concerns our work.

---

[2]Since these projections are of the same size and no other projection has the same size, transitions are allowed only between them in our current implementation.

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

CONTENTS

## A CONVERGENCE PROOF OF EVOPRESS

### A.1 A WARM-UP ARGUMENT FOR SINGLE OFFSPRING

The overall goal of this section is to prove Theorem 1. As the main argument is quite complex, relying heavily on stochastic drift analysis, we begin with a warm-up, namely by presenting a simpler proof for the restricted case where $\lambda = 1$.

Unlike the practical application of Algorithm 1, this section assumes that each fitness evaluation returns the exact, or 'true,' fitness value, ignoring any noise introduced by minibatching. Additionally, our results hold for any initialization. To align with standard notation in the runtime analysis of evolutionary algorithms, we will count generations starting from zero (i.e., using 0-based indexing).

**Theorem 2** (Single offspring). *Let $n, k \in \mathbb{N}$ with $k \leq n$ and consider the $(1+1)$-EA with level-switch mutation. Then any linear fitness function $f : \{x \mid x \in \{0,1\}^n, |x|_1 = n - k\} \to \mathbb{R}$ is optimized in expected*

$$O(k \cdot (n - k)) \text{ generations.}$$

*Proof.* Let $w \in \mathbb{R}^n$ be the weights associated to the linear function such that $f(x) = \sum_{i=1}^n x_i \cdot w_i$. To derive an upper bound we can assume that no two weights are equal [3]. Furthermore, assume without loss of generality that these weights are sorted increasingly, meaning $w_1 < w_2 < ... < w_n$, and that $k \leq (n - k)$, as the other case follows from symmetry. Since $f$ is defined on the bit strings with exactly $k$ 0's its unique optimum is now given by $x_{\text{opt}} = 0^k 1^{n-k}$. Denote by $x^{(t)}$ the search point at step $t$ and let

$$T = \inf\{t \geq 0 \mid x^{(t)} = x_{\text{opt}}\}$$

be the number of generations required until the optimum is found.

Define $X^{(t)} = \sum_{j=1}^k x_j^{(t)}$ as the random variable that captures the number of 1's in the first $k$ bits of the search point at step $t$. We observe the following:

1. $X^{(t)} = 0 \Leftrightarrow x^{(t)} = x_{\text{opt}}$;

2. $X^{(t)}$ is non-increasing;

3. $X^{(t)} - X^{(t+1)} \leq 1$;

4. $X^{(0)} = \sum_{j=1}^k x_j^{(0)}$.

It follows that given the initial search point $x^{(0)}$ we can decompose $T$ into $s = \sum_{j=1}^k x_j^{(0)}$ stages $T_1, T_2, ..., T_s$, where $T_j = \inf(\{t \geq 0 \mid X^{(t)} = j - 1\}) - \inf(\{t \geq 0 \mid X^{(t)} = j\})$ captures the number of generations spent at stage $j$. By linearity of expectation we have

$$\mathbb{E}[T \mid X^{(0)} = s] = \sum_{j=1}^s \mathbb{E}[T_j].$$

It remains to bound the expected time spent at each stage. Each offspring is generated by copying the parent, selecting a 1-bit uniformly at random, selecting a 0-bit uniformly at random and finally flipping both bits. At stage $j$ exactly $j$ of the $k$ 0-bits are among the last $n - k$ positions and exactly $j$ of the $n - k$ 1-bits are among the $k$ first positions. Hence, $j^2$ out of the total $k(n - k)$ (1-bit position, 0-bit position)-pairs advance the optimization to the next stage, yielding

$$\mathbb{P}[X^{(t+1)} = j - 1 \mid X^{(t)} = j] = \frac{j^2}{k(n - k)}.$$

Therefore, $T_j \sim \text{Geometric}(\frac{j^2}{k(n-k)})$ and

$$\mathbb{E}[T_j] = \frac{k(n - k)}{j^2}.$$

---

[3]Formally, this can be shown using stochastic domination, which involves coupling the potentials in both cases and proving that, given the same randomness, one is always at least as large as the other.

To obtain an upper bound, we can make a worst-case assumption by setting $X(x^{(0)}) = k$. We conclude

$$\mathbb{E}[T] \leq \mathbb{E}[T|X^{(0)} = k] = \sum_{j=1}^{k} \mathbb{E}[T_j] = k(n-k) \sum_{j=1}^{k} \frac{1}{j^2} \in O(k(n-k)).$$

$\square$

**Discussion.** Observe that, under the assumption that the probability of initializing at the optimum is sufficiently small, the proof is tight up to a constant factor of 2.

It is important to note that the above proof relies on the key assumption that whenever one of the $j^2$ "good" pairs is selected during mutation, the resulting offspring is the fittest among all candidates. This condition holds naturally when there is only a single offspring, as the offspring produced by flipping one of the $j^2$ pairs will have higher fitness than the parent. However, in the case of multiple offspring, this approach breaks down, as an offspring produced by flipping one of the $j^2$ "good" pairs might still have lower fitness than another offspring that was not generated by flipping one of these $j^2$ "good" pairs.

## A.2 THE MAIN ARGUMENT

Drift analysis, originally developed to study of random walks and Markov chains, has become the most widely used technique for analyzing the runtime of evolutionary algorithms in recent years. It works by first defining a potential function $X^{(t)}$ that measures the progress over each step $t$ of the optimization. By estimating how this potential changes at each step in expectation, i.e., computing the *drift* in $X^{(t)}$, one can then make probabilistic statements about the number of steps required until the potential reaches a certain threshold, also called the *hitting time*. To this end, a variety of drift theorems have been established, two of which will be employed in our proof. For a more thorough introduction to Drift Analysis we refer to Lengler (2020).

First of all, we will utilize the the Multiplicative Drift Theorem, more specifically a tail bound introduced by Doerr and Goldberg, which is applicable when the potential decreases by a constant fraction in each step.

**Theorem 3** (Multiplicative Drift, Tail Bound (Doerr & Goldberg, 2010)). *Let $(X^{(t)})_{t \geq 0}$ be a sequence of non-negative random variables over a finite state space $S \subset \mathbb{R}_0^+$. Assume that $X^{(0)} \leq b$ and let $T$ be the random variable that denotes the first point in time $t \in \mathbb{N}$ for which $X^{(t)} \leq a$, for some $a \leq b$. Suppose that there exists $\delta > 0$ such that for all $t < T$,*

$$\mathbb{E}[X^{(t)} - X^{(t+1)} \mid X^{(t)}] \geq \delta X^{(t)}$$

*Then,*

$$\mathbb{P}[T > \frac{t + \log(b/a)}{\delta}] \leq e^{-t}.$$

Additionally, we will employ Johannsen's Variable Drift Theorem. This theorem provides more flexibility compared to the Multiplicative Drift Theorem, as it can be applied when the drift is bounded by any *increasing* function of the potential. This often occurs naturally, as optimization typically becomes more difficult approaching the optimum.

**Theorem 4** (Variable Drift Theorem (Johannsen, 2010)). *Let $(X^{(t)})_{t \geq 0}$ be a sequence of non-negative random variables over a finite state space $S \subset \mathbb{R}_0^+$. Let $s_{min} := \min(S \setminus \{0\})$, let $T := \inf\{t \geq 0 \mid X^{(t)} = 0\}$, and for $s \in S$ let $\Delta^{(t)}(s) := \mathbb{E}[X^{(t)} - X^{(t+1)} \mid X^{(t)} = s]$. If there is an increasing function $h : \mathbb{R}^+ \to \mathbb{R}^+$ such that for all $s \in S \setminus \{0\}$ and all $t \geq 0$,*

$$\Delta^{(t)}(s) \geq h(s),$$

*then*

$$\mathbb{E}[T] \leq \frac{s_{min}}{h(s_{min})} + \mathbb{E}\left[\int_{s_{min}}^{X^{(0)}} \frac{1}{h(\sigma)} d\sigma\right],$$

*where the expectation on the latter term is over the random choice of $X^{(0)}$.*

We will first prove an auxiliary Lemma, which will play a central role for bounding the drift. For this purpose, we define an *inversion* in a bit string $x \in \{0,1\}^n$ as a pair of indices $(i,j)$ such that $i < j$ and $x_i > x_j$. The distance between these indices, $j - i$, will be referred to as the *spread* of this inversion.

**Lemma 1.** *Let $x \in \{0,1\}^n$ be an arbitrary bit string with $k$ 0-bits and denote by $s$ the number of inversions in $x$. Then, the average spread of these inversions is at least $\sqrt{s}/16$.*

*Proof.* Consider the bit string $1^{n-k}$ containing all 1-bits of $x$. We can now generate an arbitrary bit string $x \in \{0,1\}^n$ with $k$ 0-bits and $s$ inversions by adding $k$ 0-bits in such a way that $s$ inversions are generated. Observe that adding a 0-bit after the $j$'th 1-bit results in exactly $j$ additional inversions, regardless of the other 0-bits. This means that the order in which the 0-bits are added does not effect the outcome. We proceed by a case distinction depending on how the inversions are generated.

**Case 1:** at least $s/2$ inversions are generated by adding 0-bits after the $\sqrt{s}$'th 1-bit.

For each 0-bit that is added after the $\sqrt{s}$'th 1-bit, at least half of the resulting inversions have spread at least $\sqrt{s}/2$. Consequently, this implies that there are at least $s/4$ inversions having spread at least $\sqrt{s}/2$ in total.

**Case 2:** fewer than $s/2$ inversions are generated by adding 0-bits after the $\sqrt{s}$'th 1-bit.

It follow that more than $s/2$ inversions are generated by adding 0-bits not after the $\min(n-k, \sqrt{s})$'th 1-bit. Observe that each 1-bit can participate in at most $j$ inversions with spread at most $j$. More specifically, each 1-bit can be part of at most $\sqrt{s}/4$ inversions with spread at most $\sqrt{s}/4$. Because all of the $s/2$ inversions that are added contain one of the first $\min(n-k, \sqrt{s})$ 1-bits, at most $s/4$ of these inversions can have spread at most $\sqrt{s}/4$. Therefore, we conclude that the average spread of all inversions must be at least $\sqrt{s}/16$. $\qquad\square$

We continue to prove the final result.

**Proof of Theorem 1**

*Proof.* As in the proof of Theorem 2 let $w \in \mathbb{R}^n$ represent the weights associated with a linear function of the form $f(x) = \sum_{i=1}^n x_i \cdot w_i$. To establish an upper bound, we can again assume that no two weights are equal. Additionally, without loss of generality, assume that the weights are ordered in increasing value, i.e., $w_1 < w_2 < \cdots < w_n$, and that $k \le n-k$, as the other case follows by symmetry. Let $x^{(t)}$ denote the search point at step $t$, and define

$$T = \inf\{t \ge 0 \mid x^{(t)} = 0^k 1^{n-k}\}$$

as the number of generations required to reach the optimal solution.
Consider the potential function

$$X^{(t)} = \sum_{i=1}^n (1 - x_i^{(t)}) \cdot i - \frac{k \cdot (k+1)}{2},$$

which captures the number of inversions at step $t$. Since $x_{\text{opt}} = 0^k 1^{n-k}$ is the only bit string with $k$ 0-bits without inversions, we have $X^{(t)} = 0$ if and only if $x^{(t)} = x_{\text{opt}}$. At the same time, no bit string with $k$ 0-bits has more than $k(n-k)$ inversions, hence, $X^{(t)} \le k(n-k)$ at all times. During mutation, each of the $\lambda$ offspring is generated independently by copying the parent $x^{(t)}$, choosing uniformly at random one of the 1-bits, choosing uniformly at random one of the 0-bits and finally flipping both bits. This flipping can also be viewed as switching both bits, so that bits "move" across the search point in consecutive generations. We will use this abstraction in a later step of the proof.

As we assume the weights to be ordered increasingly, an offspring is fitter than its parent if and only if the chosen 1-bit was to the left of the chosen 0-bit, meaning, the chosen pair during mutation was

an inversion. Since there are $k(n-k)$ possible pairs in total, we have for each offspring $y_1, ..., y_\lambda$

$$\mathbb{P}[f(y_j) > f(x^{(t)}) \mid X^{(t)} = s] = \frac{s}{k(n-k)}.$$

At the same time, switching two bits corresponding to an inversion decreases the number of inversions by the difference in their positions, which we call the *spread* of an inversion. This implies that any offspring fitter than its parent must have fewer inversions than its parent and therefore, $X^{(t+1)} \leq X^{(t)}$ for all $t$. Note that we cannot make the same statement about the entire group of offspring, meaning, the fittest offspring is not guaranteed to have the fewest inversions. Since $X^{(t)}$ is non-increasing we can decompose $T$ into the number of steps required until for the first time the current search point $x^{(t)}$ has at most $\frac{k(n-k)}{\lambda}$ inversions and the number of steps required from there until the optimum is found. By linearity of expectation

$$\mathbb{E}[T] = \mathbb{E}[T_1] + \mathbb{E}[T_2],$$

where

$$T_1 = \inf\{t \geq 0 \mid X^{(t)} \leq \frac{k(n-k)}{\lambda}\}$$

and

$$T_2 = \inf\{t \geq 0 \mid X^{(t)} = 0\} - \inf\{t \geq 0 \mid X^{(t)} \leq \frac{k(n-k)}{\lambda}\}.$$

In the remainder of this proof we will demonstrate that each of these two phases requires only an expected $O(k(n-k)/\lambda)$ generations.

We begin by bounding the expected number of steps until the search point has at most $k(n-k)/\lambda$ inversions. As computed previously, a single offspring is fitter than its parent with probability $\frac{s}{k(n-k)}$. Since any fitter offspring has fewer inversion than its parent, the potential decreases in a given step, if and only if, at least one of the offspring is fitter. By using that each offspring is generated independently and that $s \geq \frac{k(n-k)}{\lambda}$ for this phase we get that

$$\mathbb{P}[X^{(t+1)} < X^{(t)} \mid X^{(t)} = s] = 1 - (1 - \frac{s}{k(n-k)})^\lambda \geq 1 - e^{\frac{-\lambda s}{k(n-k)}} \geq 1 - e^{-1}.$$

This means, in phase 1 we have a constant probability of decreasing the potential every step. However, the resulting constant drift only provides an upper bound of $O(k(n-k))$ via the Additive Drift Theorem (He & Yao, 2004). Improving this constant drift bound is challenging because we must establish a lower bound on the expected reduction in the number of inversions, given the existence of a fitter offspring. The number of inversions in an offspring is not independent of its fitness, and there is no guarantee that a fitter offspring will have fewer inversions than a less fit one. This issue is mitigated when there is only a single fitter offspring (as demonstrated in the proof of phase 2), but it becomes problematic when multiple offspring are fitter than the parent with high probability. For example, consider the bit string $1^1 0^{10} 1^{100} 0^1$ with corresponding weights $w_1 = 1, w_2 = 1002, w_3 = 1003, ..., w_{112} = 1112$. If $\lambda$ is reasonably large it becomes very likely that at least one of the children will have the first 1-bit chosen in mutation. This offspring is guaranteed to be the fittest one, but at the same time (assuming the chosen 0-bit is not the last one) it decreases the number of inversions very little compared to sampling one of the inversions for mutation uniformly at random. We will resolve this difficulty by a separate drift argument.

Let $B_C$ be the event that, within the next

$$\frac{2C}{1 - e^{-1}} \frac{k(n-k)}{\lambda}$$

steps, the number of inversions in $x^{(t)}$ falls below the threshold of

$$\frac{k(n-k)}{\lambda}.$$

Here, $C$ is chosen such that $\lambda \leq \frac{C}{4} \frac{n}{\log(n)}$. If we can demonstrate that $B_C$ occurs with a probability of at least some constant, then the proof of the first phase is established, as $B_C$ is expected to occur after a constant number of repetitions.

Henceforth, we will implicitly condition on $s \geq k(n-k)/\lambda$, since otherwise, the conclusion follows immediately. By the Chernoff bound over round events, the probability that the potential decreases at most

$$C\frac{k(n-k)}{\lambda}$$

times within the next

$$\frac{2C}{1-e^{-1}}\frac{k(n-k)}{\lambda}$$

rounds is sub-constant. We will condition on the event that the potential decreases at least

$$C\frac{k(n-k)}{\lambda}$$

times, and from now on, we will only consider such potential-reducing generations.

If we regard mutation as swapping the 1-bit with the 0-bit, we can enumerate all 0-bits from 1 to $k$ and denote by $i_j$ the current position of the $j$'th 0-bit, which will be referred to as $0_j$. Note that this enumeration stays fixed across generations, meaning that the relative order can change and $0_j$ is not necessarily the $j$'th 0-bit in $x^{(t)}$. Now define

$$Z_j^{(t)} = 1 + \sum_{l=1}^{i_j} x^t$$

as the random variable that captures the number of 1-bits before $0_j$ plus one, or in other words, one plus the number of inversions this specific 0-bit is part of. Let $S_j$ denote the event that the fittest offspring was generated by a mutation that selected $0_j$ and this offspring is fitter than the parent. We continue to show that

$$\mathbb{E}[Z_j^{(t+1)} \mid Z_j^{(t)} = s, S_j] = \frac{s}{2}.$$

We achieve this by systematically revealing the randomness in each generation. First, uncover which 0-bit flip produced the fittest offspring[4]. Assume this bit is $0_j$. Next, reveal all offspring that were generated by flipping other 0-bits than $0_j$. Let $m$ be the number of offspring that were not uncovered yet, i.e., the number of offspring where $0_j$ was switched. Now enumerate all 1-bits left of $0_j$ in $x^{(t)}$ from right to left (here, relative order matters). Let $l$ be the smallest integer such that when switching the $l$'th 1-bit left of $0_j$ with $0_j$ the resulting offspring of $x^{(t)}$ has higher fitness than all $\lambda - m$ previously uncovered offspring. Denote by $D_l$ the corresponding event. Such $l$ must exists, since we condition on the event that some offspring with bit $0_j$ flipped (switched) is the fittest among all offspring. Because the weights are sorted increasingly it must hold that switching the $l + 1$'th 1-bit with $0_j$ will also result in an offspring with higher fitness than the other $\lambda - m$ offspring, while switching the $l - 1$'th 1-bit with $0_j$ will result in an offspring with lower fitness than the other $\lambda - m$ offspring. Next, uncover all offspring where bit $0_j$ was switched with one the first up to $(l-1)$'th bit left of $0_j$. Let $m'$ denote the number of yet uncovered offspring. Now each of the remaining $m'$ offspring is generated by flipping $0_j$ with one of the $l$'th to $s$'th 1-bits left of $0_j$. Observe that the fittest among them will be the one with the leftest 1-bit chosen. Therefore,

$$\mathbb{E}[Z_j^{(t+1)} \mid Z_j^{(t)} = s, S_j, D_l, m' \text{ offspring not uncovered}] = s - \mathbb{E}\left[\max_{i=1,\ldots,m'} U_i\right],$$

where $U_i \sim \text{Uniform}(l, s)$. Given that we are conditioning on $S_j$, we know that the fittest offspring was produced by flipping $0_j$, which implies $m' \geq 1$. It follows that

$$\mathbb{E}[Z_j^{(t+1)} \mid Z_j^{(t)} = s, \ S_j] \geq s/2.$$

Denote by $\hat{T}_j$ the number of steps required until $Z_j$ reaches 1, only counting steps where $Z_j$ is decreased. Using a tail bound for the Multiplicative Drift Theorem (Theorem 3) we have that

$$\mathbb{P}[\hat{T}_j > 2(\log(n) + \log(n-k))] \leq \frac{1}{n}.$$

---

[4]To be more precise, we must uncover which 0-bit flip produced the offspring chosen during selection, to account for the case that multiple offspring have the same highest fitness (in case of a draw, one usually samples one of the fittest candidates uniformly at random). Since the case of multiple offspring with identical fitness is a mere technicality, we have largely omitted it.

As $k < (n-k)$ we conclude by a union bound that with probability at least $1/2$ each potential $Z_j$ will reach 1 within at most $2\log(n)$ steps. Therefore, with probability at least $1/2$, after $2k\log(n)$ generations where some offspring is fitter than the parent, there must be 0 inversions in $x^{(t)}$. However, note that in practice, there will not actually be 0 inversions in $x_t$, as the condition $s \geq k(n-k)/\lambda$ is violated earlier, leading the optimization process to enter the second phase. Using the fact that $\lambda \leq \frac{C}{4}\frac{n}{\log(n)}$ and $n - k \geq n/2$ we obtain

$$2k\log(n) \leq \frac{4k(n-k)\log(n)}{n} \leq C\frac{k(n-k)}{\lambda}.$$

Finally, as the probability of having less than $Ck(n-k)/\lambda$ "successful" generations in the considered time period is sub-constant, we conclude via another union bound that there exists a constant $C'$ such that event $B_C$ occurs with probability at least $1/C'$. Consequently, we have

$$\mathbb{E}[T_1] \leq C \cdot C' \cdot \frac{k(n-k)}{\lambda} \in O\left(\frac{k(n-k)}{\lambda}\right).$$

To compute $\mathbb{E}[T_2]$ we first bound the probability that exactly one of the generated offspring is fitter than the parent. Denote by

$$A_i = \left\{\left|\left\{j \in \{1, \ldots, \lambda\} \mid f(y_j) > f(x^{(t)})\right\}\right| = i\right\}$$

the event that exactly $i$ of the offspring are fitter than the parent $x^{(t)}$. As shown earlier, the probability that a given offspring is fitter than its parent is exactly $\frac{s}{k(n-k)}$, where $s$ represents the number of inversions in $x^{(t)}$. Given that each offspring is generated independently, we have for $s \leq \frac{k(n-k)}{\lambda}$

$$\mathbb{P}[A_1 \mid X^{(t)} = s] = \lambda \cdot \frac{s}{k(n-k)} \cdot \left(1 - \frac{s}{k(n-k)}\right)^{\lambda-1}$$

$$\geq \lambda \cdot \frac{s}{k(n-k)} \cdot \left(1 - \frac{s}{k(n-k)}\right)^{\frac{k(n-k)}{s}-1}$$

$$\geq \lambda \cdot \frac{s}{k(n-k)} \cdot \frac{1}{e}.$$

Lemma 1 indicates that when selecting an offspring uniformly at random from all those with higher fitness than the parent (i.e., those generated by flipping an inversion), the expected number of inversions in that offspring is at least $\sqrt{s}/16$ fewer than in the parent. We can now reveal the randomness in two steps. First, we only uncover how many of the generated offspring are fitter than the parent. Given that there is only a single fitter offspring, i.e., conditioned on $A_1$, we then uncover its number of inversions. Clearly, this single fitter offspring is now sampled uniformly at random from all offspring with higher fitness than $x^{(t)}$; thus, for $s \leq \frac{k(n-k)}{\lambda}$

$$\Delta^{(t)}(s) = \mathbb{E}[X^{(t+1)} - X^{(t)} \mid X^{(t)} = s]$$

$$= \sum_{k=0}^{\lambda} \mathbb{E}[X^{(t+1)} - X^{(t)} \mid X^{(t)} = s, A_k] \cdot \mathbb{P}[A_k \mid X^{(t)} = s]$$

$$\geq \mathbb{E}[X^{(t+1)} - X^{(t)} \mid X^{(t)} = s, A_1] \cdot \mathbb{P}[A_1 \mid X^{(t)} = s]$$

$$\geq \frac{\sqrt{s}}{16} \cdot \lambda \cdot \frac{s}{k(n-k)} \cdot \frac{1}{e}.$$

Finally, applying Johannsen's Variable Drift Theorem (Johannsen, 2010) (Theorem 4) yields

$$\mathbb{E}[T_2] \leq 16e\frac{k(n-k)}{\lambda} + \mathbb{E}\left[\int_1^{X^{(0)}} 16e\frac{k(n-k)}{\lambda\sigma^{3/2}}\,d\sigma\right]$$

$$\leq 16e\frac{k(n-k)}{\lambda}\left(1 + \int_1^{\frac{k(n-k)}{\lambda}} \frac{1}{\sigma^{3/2}}\,d\sigma\right)$$

$$\in O\left(\frac{k(n-k)}{\lambda}\right).$$

$\square$

# B  EVOLUTIONARY SEARCH PARAMETER ABLATIONS

## B.1  MUTATATION RATE (DEPTH PRUNING)

The mutation rate plays a crucial role in balancing exploration and exploitation. A higher mutation rate allows for broader exploration of the search space; however, this space grows exponentially with the number of mutations. As a result, when attempting to approach the optimum in terms of Hamming distance, the proportion of "good" offspring decreases significantly with increasing mutation rates. Consequently, in a smooth fitness landscape, we expect faster optimization with a lower mutation rate. To study the impact of mutation rate on our search process, we tested various distributions from which the number of mutations is sampled. Table 4 illustrates the effects of these distributions on the task of selecting the optimal 12 blocks to drop for Mistral-7B-v0.3. The results confirm our intuition: higher mutation rates generally reduce performance. However, sampling from the minimum of two uniform distributions ensures a reasonably high probability of selecting a low number of mutations. These offspring, with fewer mutations, then drive the optimization process, yielding to comparably lower performance drops. Conversely, when we eliminate this sampling and instead use a high, constant mutation rate, we lose the locality that is crucial for evolutionary algorithms, leading to a significant drop in performance.

Table 4: Effect of varying the distribution determining the number of mutations.

| Number of Mutations | | Wiki2↓ | C4↓ | FW↓ |
|---|---|---|---|---|
| $\min(U_1, U_2)$, | $U_1, U_2 \sim U(1,3)$ | **17.52** | 21.60 | 16.79 |
| $\min(U_1, U_2)$, | $U_1, U_2 \sim U(1,7)$ | 21.49 | 22.41 | 17.65 |
| $\min(U_1, U_2)$, | $U_1, U_2 \sim U(1,15)$ | 18.65 | 22.67 | 17.63 |
| 1 | | 18.12 | **21.12** | **16.33** |
| 3 | | 22.09 | 25.42 | 19.25 |
| 7 | | 25.06 | 26.52 | 19.65 |
| 15 | | 27.01 | 28.19 | 22.03 |

A low mutation rate carries the risk of getting trapped in local optima. However, as discussed in Section 3, we expect the dynamic model compression problem to exhibit a smooth fitness landscape with few local optima. Moreover, fitness evaluations in our context are relatively expensive. Increasing the mutation rate would only be beneficial if the smaller search space had already been thoroughly explored. In our case, though, even a small neighborhood of the search space cannot be fully explored within a feasible time frame.

A widely used strategy for balancing the advantages and disadvantages of different mutation rates involves self-adjusting mutation rates, which have been shown to be effective both theoretically and in practice (Kern et al., 2004; Doerr et al., 2019). These methods decrease the mutation rate when progress is relatively "easy", and increase it when progress becomes difficult, offering a greater chance of escaping local optima.

B.2   MULTI-STEP SELECTION (UNSTRUCTURED SPARSITY)

We will use this subsection to ablate the impact of hyperparameters for multi-step selection, namely, the number of tokens and survivors. As discussed earlier in Section 4.2, the default hyperparameters we chose for our unstructured sparsity search were quite conservative. The following experiments will be conducted based on the super fast version, which uses two steps of selection. It first generates 16 offspring, evaluates them on 512 tokens, and compares only the fittest one with the parent on another 8192 tokens.

Table 5 shows the impact of adapting the number of tokens in the first selection step. Note that reducing tokens is only reasonable up to a certain degree, as fitness evaluation has constant overhead independent of the number of tokens (e.g., for loading the levels). Table 6 ablates the number of offspring in each generations. All perplexities were measured after 400 generations.

Table 5: Effect of varying the number of tokens in first preselection step.

| Offspring | Stage 1: Tokens | Stage 2: Tokens | Wiki2↓ | C4↓ | FW↓ |
|---|---|---|---|---|---|
| 16 | 1024 | 8192 | 16.22 | **17.93** | **12.26** |
| 16 | 12 | 8192 | **15.87** | 18.28 | 12.38 |
| 16 | 256 | 8192 | 17.25 | 18.51 | 12.52 |
| 16 | 128 | 8192 | 16.01 | 18.99 | 12.72 |
| 16 | 64 | 8192 | 15.89 | 19.35 | 12.98 |

Table 6: Effect of varying the number of offspring.

| Offspring | Stage 1: Tokens | Stage 2: Tokens | Wiki2↓ | C4↓ | FW↓ |
|---|---|---|---|---|---|
| 64 | 512 | 8192 | 16.35 | 18.27 | **12.36** |
| 32 | 512 | 8192 | 16.65 | **18.22** | 12.44 |
| 16 | 512 | 8192 | **15.87** | 18.27 | 12.38 |
| 8 | 512 | 8192 | 16.37 | 18.74 | 12.64 |
| 4 | 512 | 8192 | 17.87 | 18.97 | 12.72 |

In a similar vein to the discussion in Appendix B.1, the number of offspring can also be dynamically adapted. Ideally, the number of offspring should increase to the point where the computational effort is compensated by the number of fitness evaluations, as outlined in Theorem 1. Methods such as the Self-Adjusting $(1, \lambda)$-EA have recently gained significant theoretical interest and have been shown to automatically determine "ideal" offspring sizes on specific problems (Hevia Fajardo & Sudholt, 2021; Kaufmann et al., 2022). Although we have not experimented with such adaptive methods, we see significant potential for future work in this area, particularly considering the multi-step selection we employ.

B.3   FITNESS ENVIRONMENT (QUANTIZATION)

We explored alternative fitness functions by testing perplexity as opposed to KL-Divergence. One advantage of using perplexity is the reduced memory requirement, as it does not necessitate storing logits, which can be particularly burdensome for large vocabularies. However, perplexity relies solely on the information from the ground truth token, while KL-Divergence takes into account the entire distribution. This distinction is significant only if the selection decisions vary between the two metrics. Generally, we expect KL-Divergence to perform at least as well as perplexity; however, in many instances, their performances are similar. This observation could indicate that KL-Divergence might be using more tokens than necessary to assess fitness effectively. Although in the context of quantization, KL-Divergence yielded slightly better results (Table 7), both metrics showed comparable performance when applied to unstructured sparsity (Figure 4).

Table 7: Comparison of using KL-Divergence vs. Perplexity as fitness function.

| Model | # Bits | Method | Wiki2↓ | C4↓ | FW↓ |
|---|---|---|---|---|---|
| Llama-3-8B | 3 | Uniform | 12.19 | 15.76 | 11.47 |
| | | EvoPress (PPL) | 8.17 | 12.15 | 9.64 |
| | | EvoPress (KL) | **7.49** | **12.03** | **9.56** |
| | 4 | Uniform | 6.48 | 9.50 | 8.46 |
| | | EvoPress (PPL) | **5.86** | 9.46 | 8.23 |
| | | EvoPress (KL) | **5.86** | **9.44** | **8.22** |
| Llama-2-7B | 3 | Uniform | 6.16 | 7.96 | 6.86 |
| | | EvoPress (PPL) | 5.74 | 7.90 | 6.79 |
| | | EvoPress (KL) | **5.70** | **7.87** | **6.76** |
| | 4 | Uniform | 5.48 | 7.10 | 6.40 |
| | | EvoPress (PPL) | 5.25 | 7.09 | 6.37 |
| | | EvoPress (KL) | **5.22** | **7.07** | **6.34** |
| Mistral-7B-v0.3 | 3 | Uniform | 5.54 | 8.57 | 6.96 |
| | | EvoPress (PPL) | 5.23 | 8.45 | 6.87 |
| | | EvoPress (KL) | **5.21** | **8.42** | **6.86** |
| | 4 | Uniform | 5.10 | 7.87 | 6.50 |
| | | EvoPress (PPL) | 4.85 | 7.86 | 6.49 |
| | | EvoPress (KL) | **4.84** | **7.84** | **6.48** |

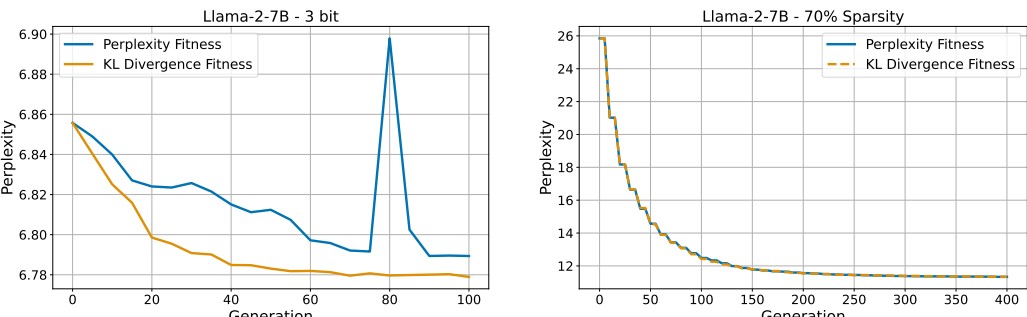

Figure 4: Convergence of EvoPress for unstructured sparsity (Left) and quantization (Right) for different fitness functions.

## C    EXPERIMENTAL SETUP

### C.1    HYPERPARAMETER SETTING

Here, we provide an overview of the hyperparameters used in our experiments. As shown in Table 8, different hyperparameters were employed for different applications due to the varying nature of their search spaces. Across all applications, we sampled the number of mutations from the distributions $\min(U_1, U_2)$ with $U_1, U_2 \sim Unif(1,3)$, which closely mimics the behavior of using only one mutation (see the ablation study in Appendix 4).

For *Depth Pruning*, where each block has only two choices and significantly fewer blocks are present compared to layers in other methods, we leveraged the insight from Theorem 1, which suggests that the number of required generations scales proportionally to $k(n-k)$, where $k$ represents the number of removed blocks and $n$ the total number of blocks.

For *Unstructured Sparsity*, the search space is considerably larger, with more than 10 choices per layer[5]. As a result, more generations are necessary to converge because each generation only makes small improvement in terms of Hamming distance from the optimum.

---

[5]If needed, one could increase the step size and reduce the number of compression levels to load.

For *Quantization*, the search space is somewhat smaller since fewer "natural" compression levels are available. However, the fitness landscape is less smooth, with significantly larger step sizes in compression levels, motivating the use of a higher number of tokens.

For all these applications, we adopted a conservative approach to the number of generations to better understand convergence. In practice, we need significantly fewer generations to converge close to optimum, as demonstrated in Section 4.2, Appendix A, and Appendix B.3. Additionally, we showed a much faster version (in terms of time per iteration) that uses significantly less tokens.

Table 8: Employed hyperparameters for different applications.

| Application | Generations | Offspring | Survivors (1) | Tokens (1) | Survivors (2) | Tokens (2) | Survivors (3) | Tokens (3) |
|---|---|---|---|---|---|---|---|---|
| Depth Pruning | $k(n-k)/1.5$ | 32 | 2 | 2048 | 1 | 32768 | N/A | N/A |
| Unstr. Sparsity | 400 | 64 | 8 | 2048 | 2 | 16384 | 1 | 65536 |
| Quantization | 150 | 128 | 16 | 2048 | 4 | 16384 | 1 | 131072 |
| Super-Fast | 400 | 16 | 1 | 512 | 1 | 8192 | N/A | N/A |

## C.2 ROBUSTNESS TO RANDOM SEED

To evaluate the robustness of EvoPress, we conducted 16 independent runs with different random seeds. Specifically, we used the "super-fast" variant to determine the optimal compression allocation for Llama-3-8B at 70% sparsity, assessing perplexity scores on the C4, Wikitext2, and hold-out Fineweb-Edu datasets. The results indicate that EvoPress is highly robust, as reflected by the low standard deviation observed across the hold-out metrics (Figure 5). For example, after 1000 generations of the "super-fast" variant, the configurations found achieve a mean C4 perplexity of 33.82 with a standard deviation of 0.61, compared to 52.32 for the next best method, OWL, highlighting the statistically significant improvements achieved by EvoPress. Furthermore, as shown in Figure 6, the configurations identified across different runs demonstrate high similarity, which is expected to improve further with additional generations.

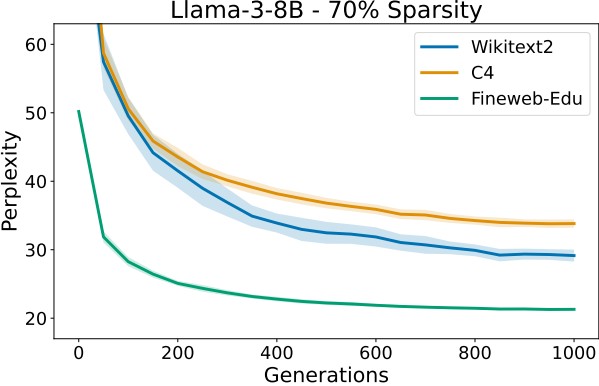

Figure 5: Convergence behavior of the "super-fast" variant across 16 independent runs. The extremely low standard deviation (shaded area) underscores the robustness of the method, suggesting that local optima do not pose significant challenges to the search.

## D ADDITIONAL DEPTH PRUNING RESULTS

### D.1 FULL PERPLEXITY TABLES

Here, we present our additional results for depth pruning experiments on Llama-2-7B (Table 9), Llama-3-8B (Table 10), and Mistral-7B-v0.3 (Table 11). Across all levels of sparsities, EvoPress consistently outperforms previous methods. Additionally, Table 11 includes results where only entire transformer blocks in the EvoPress are removed, showcasing that the significant gains are not

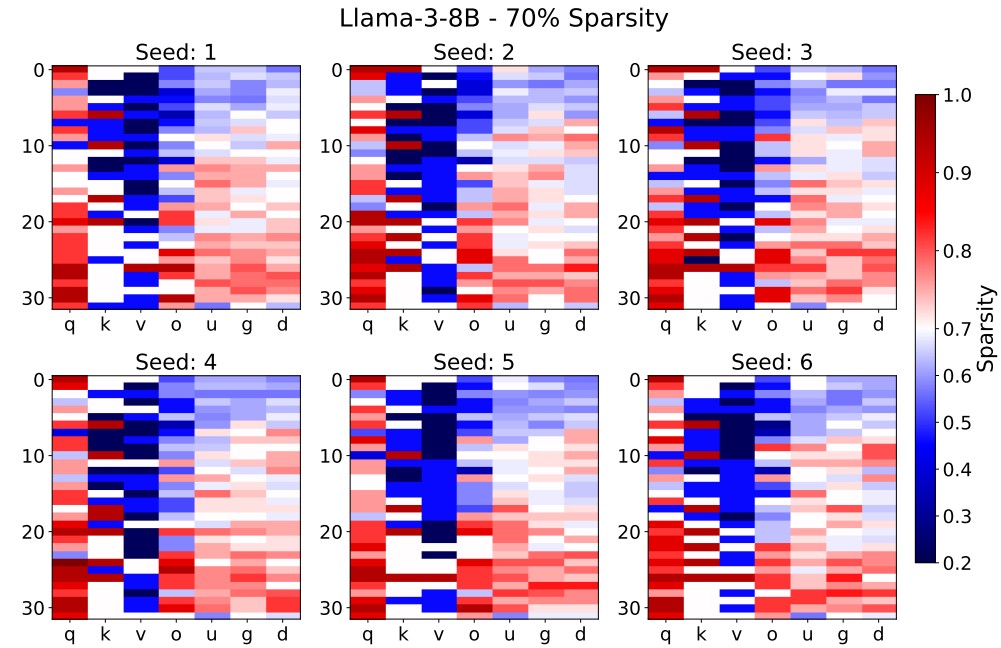

Figure 6: Configurations identified by EvoPress on Llama-3-8B after 1000 generations show high similarity across different seeds. The y-axis represents the depth of the respective transformer block, while the x-axis denotes the corresponding layer (q: query, k: key, v: value, o: output, u: MLP up, g: MLP gate, d: MLP down).

Table 9: Depth pruning of Llama-2-7B.

| Sparsity | Method | Wiki2↓ | C4↓ | FW↓ |
|---|---|---|---|---|
| 0% | Dense | 5.21 | 6.93 | 6.40 |
| 12.5% | EvoPress | **6.42** | **8.60** | **7.54** |
| | ShortGPT | 8.86 | nan | 9.48 |
| | Cosine Similarity (Window) | 7.53 | 9.82 | 8.51 |
| | Weight Subcloning | 9.09 | 11.06 | 9.60 |
| | ShortenedLlama | 7.68 | 10.44 | 8.57 |
| 25% | EvoPress | **9.15** | **11.46** | **9.69** |
| | ShortGPT | 23.41 | 30.30 | 21.16 |
| | Cosine Similarity (Window) | 16.60 | 21.04 | 17.37 |
| | Weight Subcloning | 23.41 | 30.30 | 21.16 |
| | Shortened Llama | 13.86 | 14.08 | 11.81 |
| 37.5% | EvoPress | **17.98** | **18.91** | **15.53** |
| | ShortGPT | 70.94 | 63.51 | 54.07 |
| | Cosine Similarity (Window) | 192.07 | 212.60 | 151.10 |
| | Weight Subcloning | 70.94 | 63.51 | 54.07 |
| | Shortened Llama | 35.37 | 26.07 | 20.37 |
| 50% | EvoPress | **48.84** | **42.29** | **33.57** |
| | ShortGPT | 226.14 | 171.04 | 180.51 |
| | Cosine Similarity (Window) | 4570.15 | 2876.83 | 1861.06 |
| | Weight Subcloning | 226.14 | 171.04 | 180.51 |
| | Shortened Llama | 145.78 | 87.40 | 68.79 |

primarily due to this relaxation, and that our method performs better than baselines even when dealing with this coarser search space.

Table 10: Depth pruning of Llama-3-8B.

| Sparsity | Method | Wiki2↓ | C4↓ | FW↓ |
|---|---|---|---|---|
| 0% | Dense | 5.54 | 8.80 | 7.62 |
| 12.5% | EvoPress | **7.72** | **12.61** | **10.15** |
| | ShortGPT | 13.21 | 19.56 | 14.25 |
| | Cosine Similarity (Window) | 9.54 | 14.87 | 11.64 |
| | Weight Subcloning | 13.21 | 19.56 | 14.25 |
| | Shortened Llama | 9.42 | 15.09 | 11.57 |
| 25% | EvoPress | **13.99** | **22.83** | **15.84** |
| | ShortGPT | 5527.54 | 11589.93 | 2346.13 |
| | Cosine Similarity (Window) | 5519.95 | 11629.61 | 2342.91 |
| | Weight Subcloning | 5527.54 | 11589.93 | 2346.13 |
| | Shortened Llama | 16.59 | 20.81 | 16.28 |
| 37.5% | EvoPress | **27.56** | **35.70** | **26.77** |
| | ShortGPT | 64281.36 | 13836.12 | 3789.09 |
| | Cosine Similarity (Window) | 64627.29 | 13890.14 | 3784.72 |
| | Weight Subcloning | 64381.36 | 13836.13 | 3789.09 |
| | Shortened Llama | 50.20 | 61.56 | 37.40 |
| 50% | EvoPress | **84.99** | **87.86** | **66.41** |
| | ShortGPT | 1663.97 | 1740.04 | 1588.20 |
| | Cosine Similarity (Window) | 2053.19 | 1116.47 | 694.00 |
| | Weight Subcloning | 1663.97 | 1740.04 | 1588.20 |
| | Shortened Llama | 724.86 | 666.41 | 210.30 |

Table 11: Depth pruning of Mistral-7B-v0.3.

| Sparsity | Method | Wiki2↓ | C4↓ | FW↓ |
|---|---|---|---|---|
| 0% | Dense | 4.82 | 7.72 | 6.41 |
| 12.5% | EvoPress | **6.06** | **9.00** | **7.42** |
| | EvoPress (Attn.+MLP) | 6.33 | 9.44 | 7.80 |
| | ShortGPT | 7.19 | 10.18 | 8.46 |
| | Cosine Similarity (Window) | 7.19 | 10.18 | 8.46 |
| | Weight Subcloning | 7.19 | 10.18 | 8.46 |
| | Shortened Llama | 6.64 | 9.71 | 7.94 |
| 25% | EvoPress | **8.66** | **12.04** | **9.92** |
| | EvoPress (Attn.+MLP) | 9.46 | 13.02 | 10.59 |
| | ShortGPT | 43.26 | 40.16 | 29.54 |
| | Cosine Similarity (Window) | 33.75 | 54.07 | 36.26 |
| | Weight Subcloning | 43.26 | 40.16 | 29.54 |
| | Shortened Llama | 14.94 | 19.30 | 14.73 |
| 37.5% | EvoPress | **17.52** | **21.60** | **16.90** |
| | EvoPress (Attn.+MLP) | 21.62 | 25.17 | 18.97 |
| | ShortGPT | 2898.98 | 2722.66 | 981.99 |
| | Cosine Similarity (Window) | 1034.09 | 2471.86 | 1050.56 |
| | Weight Subcloning | 2898.98 | 2722.66 | 981.99 |
| | Shortened Llama | 440.20 | 442.09 | 486.15 |
| 50% | EvoPress | **61.75** | **54.15** | **43.23** |
| | EvoPress (Attn.+MLP) | 108.91 | 99.74 | 69.07 |
| | ShortGPT | 2422.72 | 2134.92 | 1083.51 |
| | Cosine Similarity (Window) | 3411.47 | 1934.16 | 1740.91 |
| | Weight Subcloning | 2422.72 | 2134.92 | 1083.51 |
| | Shortened Llama | 5241.76 | 3595.71 | 1953.14 |

## D.2 LOCALITY OF DROPPED BLOCKS

Prior work suggests that deeper layers, excluding the final ones, are generally less effective (Gromov et al., 2024; Men et al., 2024). Figure 7 illustrates the optimal drop configurations discovered by EvoPress. While some deeper layers are indeed removed at all sparsity levels, we also observe that certain shallow layers appear to be less important. Meanwhile, the first two blocks are never

removed. However, in contrast to a heuristic proposed by Ma et al. (2023), in some case it is reasonable to remove the final block.

Table 12: First 16 blocks in removal order of ShortGPT, Weight Subcloning and Shortened Llama on three different models.

| Model | Method | Removal Order (Left to Right) |
|---|---|---|
| Llama-3-8B | ShortGPT | 25, 26, 27, 24, 28, 23, 22, 29, 20, 21, 19, 18, 30, 17, 16, 11 |
| | Weight Subcloning | 25, 27, 26, 24, 28, 23, 22, 29, 20, 21, 19, 18, 30, 17, 16, 11 |
| | Shortened Llama | 10, 08, 09, 11, 26, 25, 12, 22, 24, 23, 14, 13, 28, 06, 19, 21 |
| Llama-2-7B | ShortGPT | 27, 25, 26, 28, 29, 24, 23, 22, 21, 30, 20, 19, 18, 17, 15, 14 |
| | Weight Subcloning | 27, 25, 28, 29, 26, 24, 23, 22, 21, 19, 30, 20, 18, 17, 14, 15 |
| | Shortened Llama | 11, 12, 08, 09, 10, 06, 24, 25, 07, 14, 23, 13, 22, 21, 15, 27 |
| Mistral-7B-v0.3 | ShortGPT | 26, 25, 24, 27, 23, 22, 28, 30, 21, 29, 20, 19, 13, 17, 18, 12 |
| | Weight Subcloning | 26, 25, 24, 27, 23, 28, 22, 30, 21, 29, 20, 19, 13, 17, 12, 18 |
| | Shortened Llama | 10, 12, 13, 11, 08, 09, 14, 15, 07, 06, 04, 27, 24, 16, 25, 05 |

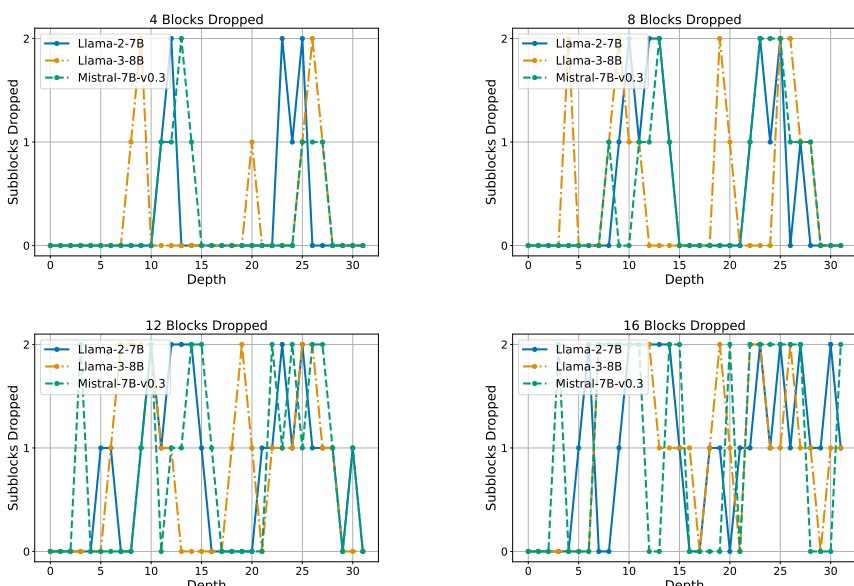

Figure 7: Optimal drop configurations produced by EvoPress for different models.

### D.3    CORRELATION OF SCORES WITH PERPLEXITY

In this experiment, we first calculated the cosine similarity and squared error for each block by comparing activations before and after the block. Next, we randomly removed subsets of blocks (excluding the first and last two) and, for each configuration, computed the average cosine similarity and squared error. The results are shown in Figure 8. Initially, the average squared error exhibited a negative correlation, as the $l2$ norm of the activations increased with depth. This lead to configurations with early blocks removed having small average error. To mitigate this, we normalized the activations prior to computing the squared error, which significantly improved the correlation, resulting in performance comparable to cosine similarity. However, as sparsity increased, the correlation degraded significantly for both methods, offering insight into why removal techniques based on scoring fail even at moderate levels of sparsity. The experiment was done using 131,072 tokens from the Fineweb-Edu calibration dataset.

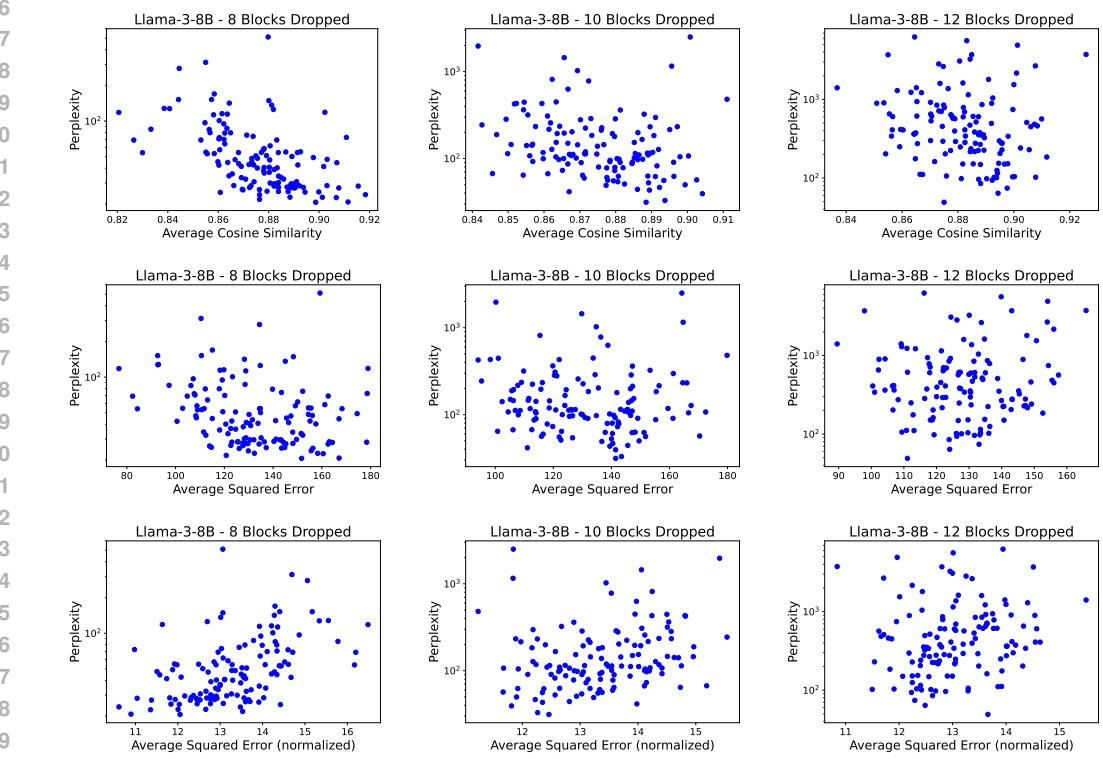

Figure 8: Effect of removing random subsets of blocks for Llama-3-8B.

# E  ADDITIONAL UNSTRUCTURED SPARSITY RESULTS

## E.1  50% AND 60% SPARSITY

In the main text, we focused on results at 70% sparsity, where the performance difference becomes more pronounced. However, since 50% and 60% sparsity levels are also commonly referenced in the literature, we also present results for these levels in Tables 13 and 14. Even at these lower sparsity levels, EvoPress demonstrates significant improvements over uniform sparsity and consistently outperforms OWL.

Table 13: Performance of various sparsity profiles at 50% sparsity

| Model | Sparsity | Wiki2↓ | C4↓ | ArcC↑ | ArcE↑ | HS↑ | PiQA↑ | WG↑ | Avg↑ |
|---|---|---|---|---|---|---|---|---|---|
| | Dense | 5.12 | 6.93 | 43.4 | 76.3 | 57.1 | 78.1 | 69.0 | 64.8 |
| Llama-2-7B | Uniform | 6.40 | 8.87 | 41.3 | 73.4 | 52.8 | 75.7 | 68.8 | 62.4 |
| | DP | 7.09 | 10.04 | 39.8 | 72.2 | 53.3 | 76.1 | 68.3 | 61.9 |
| | OWL | 6.38 | 8.77 | 41.1 | 73.2 | 53.2 | 76.5 | 70.2 | 62.9 |
| | EvoPress | **6.22** | **8.52** | 41.5 | 74.2 | 54.0 | 76.7 | 69.6 | **63.2** |
| | Dense | 5.54 | 7.10 | 50.4 | 80.1 | 60.2 | 79.7 | 72.6 | 68.6 |
| Llama-3-8B | Uniform | 8.05 | 13.07 | 43.6 | 75.7 | 54.2 | 76.1 | 71.7 | 64.3 |
| | DP | 9.45 | 14.46 | 39.8 | 72.0 | 52.9 | 74.7 | 67.2 | 61.3 |
| | OWL | 8.13 | 13.12 | 43.8 | 75.8 | 54.0 | 75.7 | 72.2 | 64.3 |
| | EvoPress | **7.63** | **12.53** | 43.9 | 77.5 | 54.5 | 76.8 | 72.2 | **65.0** |

Table 14: Performance of various sparsity profiles at 60% sparsity

| Model | Sparsity | Wiki2↓ | C4↓ | ArcC↑ | ArcE↑ | HS↑ | PiQA↑ | WG↑ | Avg↑ |
|---|---|---|---|---|---|---|---|---|---|
| | Dense | 5.12 | 6.93 | 43.4 | 76.3 | 57.1 | 78.1 | 69.0 | 64.8 |
| Llama-2-7B | Uniform | 9.3 | 12.37 | 35.8 | 69.5 | 45.9 | 72.4 | 65.9 | 57.9 |
| | DP | 15.61 | 20.73 | 32.3 | 64.6 | 43.5 | 68.5 | 63.9 | 54.6 |
| | OWL | 8.35 | 11.00 | 36.0 | 69.1 | 47.5 | 73.2 | 66.2 | 58.4 |
| | EvoPress | **8.21** | **10.34** | 37.1 | 70.6 | 49.3 | 74.4 | 67.6 | **59.8** |
| | Dense | 5.54 | 7.10 | 50.4 | 80.1 | 60.2 | 79.7 | 72.6 | 68.6 |
| Llama-3-8B | Uniform | 13.86 | 21.43 | 35.2 | 69.7 | 45.6 | 72.2 | 68.0 | 58.2 |
| | DP | 19.74 | 29.46 | 36.1 | 67.0 | 45.8 | 72.1 | 64.9 | 57.2 |
| | OWL | 12.37 | 18.53 | 38.0 | 70.3 | 47.7 | 72.1 | 68.5 | 59.3 |
| | EvoPress | **11.02** | **16.37** | 39.0 | 71.9 | 48.6 | 74.0 | 69.1 | **60.5** |

### E.2 SPARSITY PROFILES

Below, we visualize sparsity profiles determined by EvoPress and baseline approaches. It can be observed that EvoPress prunes the initial blocks less aggressively compared to the middle and later blocks, while the final block is kept relatively dense. Furthermore, the q_proj and k_proj projections achieve higher sparsity levels, whereas the o_proj and v_proj projections are pruned to lower sparsity levels on average.

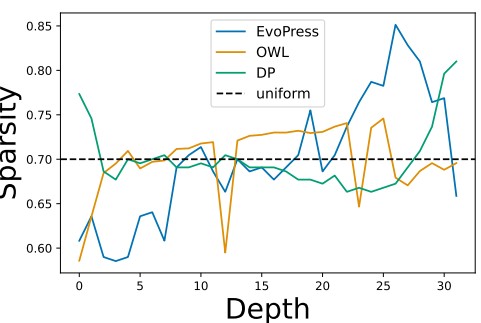

Figure 9: Block-level sparsity profiles for Llama-3-8B at 70% sparsity.

Figure 10: Average sparsity per projection type for Llama-3-8B at 70% sparsity for EvoPress.

## F ADDITIONAL QUANTIZATION RESULTS

### F.1 2.25 BIT AND 2.5 BIT

In addition to the 3 bit results presented in Section 4.3, we further evaluated EvoPress under extreme quantization conditions, specifically testing it at 2.25 bit and 2.5 bit levels. As a baseline, we generated 32 random configurations combining 2 bit and 3 bit layers and selected the best performing setup. The results, as shown in Table 15, demonstrate that EvoPress significantly outperforms this baseline, highlighting its ability to facilitate extreme quantization levels that were previously unattainable.

### F.2 PRACTICAL CONVERGENCE

Similar to unstructured sparsity, EvoPress also demonstrates rapid convergence when applied to quantization. As shown in Figure 11, the majority of improvements occur within two GPU, with full convergence achieved after approximately eight GPU hours. If needed, this optimization time could be further shortened by tuning the hyperparameters, similarly to the super-fast version for unstructured sparsity discussed in Section 4.2. However, we observed that the convergence dynamics are less smooth compared to unstructured sparsity, likely due to the limited number of quantization

Table 15: Performance of EvoPress on 2.25 bit and 2.5 bit quantization

| Model | # Bits | Method | Wiki2↓ | C4↓ | ArcC↑ | ArcE↑ | HS↑ | PiQA↑ | WG↑ | Avg↑ |
|---|---|---|---|---|---|---|---|---|---|---|
| Llama-2-7B | 2.25 | Best of 32 | 13.18 | 18.19 | 24.8 | 50.2 | 40.3 | 66.8 | 56.1 | 47.7 |
| | | EvoPress | **9.82** | **9.93** | 29.5 | 61.8 | 46.2 | 70.3 | 59.4 | **53.4** |
| | 2.5 | Best of 32 | 9.42 | 9.01 | 29.1 | 58.6 | 46.9 | 70.1 | 62.6 | 53.5 |
| | | EvoPress | **8.03** | **7.33** | 35.3 | 68.4 | 50.8 | 73.9 | 64.2 | **58.5** |
| Llama-3-8B | 2.25 | Best of 32 | 149.85 | 432.96 | 21.2 | 29.1 | 28.1 | 55.6 | 49.8 | 36.8 |
| | | EvoPress | **23.93** | **43.17** | 23.6 | 46.9 | 39.3 | 63.6 | 56.5 | **46.0** |
| | 2.5 | Best of 32 | 21.65 | 23.92 | 25.1 | 47.6 | 41.2 | 65.6 | 56.2 | 47.1 |
| | | EvoPress | **13.93** | **18.15** | 31.7 | 61.5 | 47.9 | 71.7 | 64.3 | **55.4** |
| Phi-3-Medium | 2.25 | Best of 32 | 14.20 | 18.19 | 28.9 | 46.8 | 40.0 | 61.8 | 53.1 | 46.1 |
| | | EvoPress | **10.48** | **14.60** | 36.2 | 62.0 | 46.6 | 66.2 | 55.6 | **53.3** |
| | 2.5 | Best of 32 | 8.26 | 12.65 | 40.5 | 69.3 | 50.3 | 70.9 | 61.9 | 58.6 |
| | | EvoPress | **7.12** | **11.23** | 44.1 | 75.9 | 54.1 | 73.5 | 64.6 | **62.4** |

levels available (practically only 2, 3, and 4 bit are used), resulting in a less smooth fitness landscape.

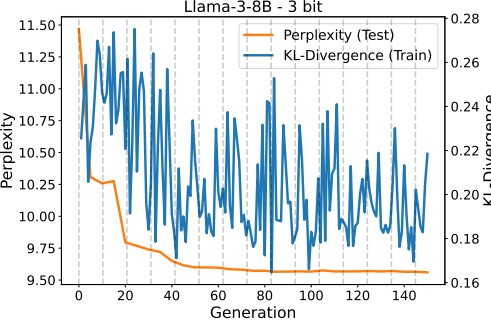

Figure 11: Convergence of EvoPress for 3 bit quantization on Llama-3-8B. Since quantization offers fewer compression levels, we observe larger changes and more instability in the training metric (KL-divergence) between steps. However, we still observe that the held-out metric (PPL) continually decreases in a smoother manner.

## F.3 DISCUSSION OF QUANTIZATION PROFILES

In this section, we visualize an quantization profile determined by EvoPress. As shown, EvoPress maintains a relatively uniform quantization bitwidth allocation across the model. However, some blocks tend to have higher bitwidth with the last one being least compressed. In addition, EvoPress transfers capacity from k_proj to v_proj.

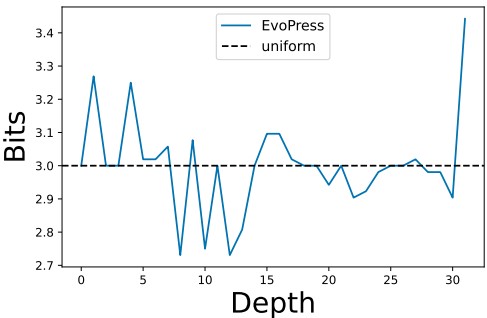 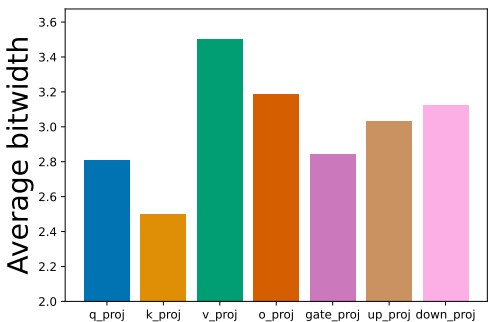

Figure 12: Block-level quantization profiles for Llama-3-8B at 3 bit compression on average.

Figure 13: Average bitwidth per projection type for Llama-3-8B at 3 bit compression on average.

## G  MULTIMODAL COMPRESSION

In the main text we considered *unimodal* compression - either depth pruning, unstructured sparsity or quantization. A natural extension of our approach is to optimize multiple compression techniques simultaneously, which we refer to as *multimodal* compression.

Below, we consider the case of joint depth pruning and quantization. To simplify the setup and search space, we apply uniform quantization to all projections within each block. The optimization process alternates between two phases:

- **Block dropping.** Multiple candidate configurations are generated by sampling blocks for removal and revival. When reintroducing a block, its weights are quantized to match the bitwidth of the removed block. The best-performing configuration is selected.

- **Quantization.** Quantization levels between "alive" blocks from the previous step are swapped, and the fittest one is retained.

Multimodal EvoPress approach yields both a set of blocks to be removed and a distribution of quantization bitwidths across the surviving blocks.

We validate the proposed approach on Llama-3.1-8B for 25% sparsity and 4-bit quantization on average (with 2, 3, 4, 5 and 6 bit options following Section 4.3). One can observe from Figure 14 that multimodal search manages to find a better solution than the starting point (the best of many uniform samples) and exhibits relatively stable convergence.

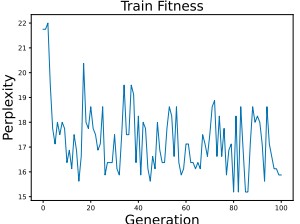 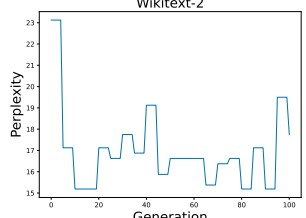 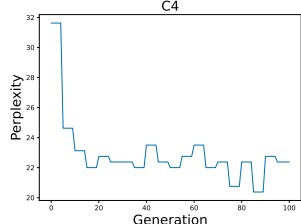

Figure 14: Convergence of multimodal EvoPress search for 25% depth pruning and 4-bit quantization on average. Perplexity on the calibration set (Left), Wikitext-2 (Middle), and C4 (Right).

# H    SCHEMATIC VISUALIZATION

We provide a schematic illustration of EvoPress in Figure 15.

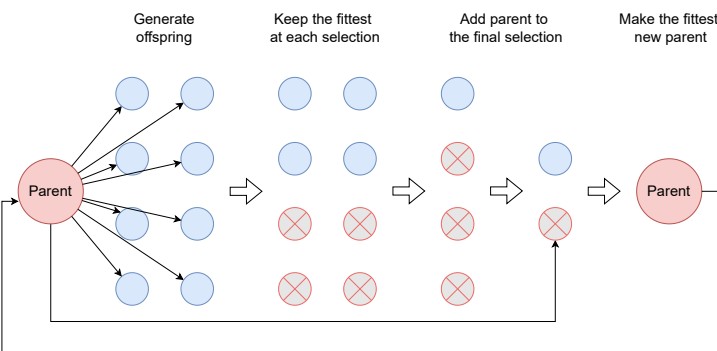

Figure 15: Schematic illustration of EvoPress search. Intially, a set of candidates is sampled. Then, a fraction of the fittest among them is selected at each elimination round. In the last selection round, the parent is added to the population for elitism. Finally, the last remaining search point is made the new parent.

