# OpenReview forum: "EvoPress: Towards Optimal Dynamic Model Compression via Evolutionary Search"
_ICLR.cc/2025/Conference — Submitted to ICLR 2025_

### Official Review · Reviewer_X4pg · 2024-11-03

**Soundness:** 3
**Presentation:** 3
**Contribution:** 2
**Rating:** 3
**Confidence:** 3

**Summary:**

This paper proposes a new evolutionary framework for dynamic LLM compression.

**Strengths:**

This paper proposes an evolutionary algorithm within an optimization framework, for sparse structure selection for LLMs.

**Weaknesses:**

1. The theoretical justification in Theorem 1 appears irrelevant to the algorithm proposed in the paper. Theorem 1 is established for linear fitness functions; however, it’s difficult to see how this applies, as LLMs likely do not have linear fitness functions.

2.  The comparison lacks systematic rigor. For instance, Table 2 presents results only at the sparsity level 70%. What about results at other sparsity levels?

**Questions:**

See weakness

---

> ### Author Response · Authors · 2024-11-18
> **Response**
>
> Thank you for your feedback. We address the two points raised below:
>
> >  “The theoretical justification in Theorem 1 is irrelevant to the algorithm proposed”
>
> We respectfully disagree with the statement that Theorem 1 is irrelevant to the algorithm.
> First, the vast majority of prior work on non-uniform compression is based on variants of this linearity assumption, either explicitly or implicitly: this is the case for ShortGPT, ShortenedLlama, Weight Subcloning, and OWL, but also dynamic-programming-based approaches such as SPDY, which makes this assumption explicit in their Equation (1). Therefore, showing _optimality_ of our algorithm under this assumption is meaningful.
>
> Second, the linearity assumption is also inherent to the widely used terminology of layer “saliency”. While we don’t argue that the fitness landscape induced by dynamic model compression is entirely linear, it follows from such standard assumptions that it behaves like a linear function to a certain degree, in particular, regarding smoothness. For example, removing the first transformer block of a model has a catastrophic effect on performance. This is the case independent of the compression allocation in other blocks, and can be regarded as assigning an importance measure to this block, resulting in a linear function to be optimized under an l1-constraint.  Third, EvoPress uses a general evolutionary algorithm, capable of optimizing a wide range of fitness landscapes. By proving that the entire class of linear functions can be optimized efficiently, we provide a theoretical foundation for the superiority of our approach compared to previous work, which does not provide any guarantees. As also noted in our paper (Footnote 1, Line 107), showing convergence on linear functions is a standard benchmark for evolutionary algorithms, see e.g. (Droste et al., 2002), (Doerr & Ku ̈nnemann, 2015), (Lengler & Spooner, 2015).
>
> Finally, we emphasize that the main contribution of this work is an empirical one. In Figure 1, we demonstrate that EvoPress is able to find the global optima for block dropping search in a constrained setting. Most importantly, we significantly outperform all previous methods–some of which are specialized to specific compression types– across a wide range of compression approaches, models, and compression levels.
>
>  > The comparison lacks rigor [...] What about results at other sparsity levels?
>
> We understand the importance of a thorough evaluation and would like to clarify that results regarding other sparsity levels _were already included in the submission_: as stated in line 434, these results for 50% are in Table 13, and for 60% in Table 14. Moreover, we would like to emphasize the thoroughness of our experiments. We cover three different compression approaches, each on three to four levels of compression, on models from three different families. By way of comparison, most previous works limited themselves to a single compression approach.

---

> > ### Author Response · Authors · 2024-12-02
> > **Discussion reminder**
> >
> > Dear Reviewer,
> >
> > As the discussion period is soon drawing to a close, we wanted to send you a gentle reminder to examine our response and let us know whether it addressed your concerns. For reference:
> >
> > * we have provided a more detailed justification of the direct relationship between Theorem 1 and our practical results, and
> > * we pointed out that the results at different sparsities were already present in our initial submission.
> >
> > Best regards,\
> > The authors

---

### Official Review · Reviewer_wUQk · 2024-11-03

**Soundness:** 3
**Presentation:** 2
**Contribution:** 2
**Rating:** 6
**Confidence:** 2

**Summary:**

This work proposes an algorithm to perform dynamic compression (different layers can be compressed to various levels) of large language models. Previous works assign importance to each layer and assume end-to-end model compression error is proportional to the sum of layer-wise errors. This work observes that error monotonicity does not hold for LLM, and proposes an evolutionary search algorithm to achieve better compression quality.

In EvoPress, multiple heuristics, including mutation operation and the selection step are proposed to achieve efficient sampling for LLM. Experimental results show better compression quality compared to previous work on multiple applications.

**Strengths:**

LLM compression is an important topic. This work proposes a new optimization algorithm, and better performance has been presented on multiple applications, including pruning, introducing sparsity, and quantization.

**Weaknesses:**

1. It would be good to introduce an algorithm overview figure to illustrate the steps in EvoPress. In addition, the algorithm contains mulitple heuristics to make the sampling efficient. It would be good to also summarize these heuristics in the figure.
2. It's not clear whether the hyperparameters used in the heuristics (mutation operation & multi-step selection) can transfer well across tasks/datasets/models. Further illustration would be very useful.

**Questions:**

Please see above

---

> ### Author Response · Authors · 2024-11-18
> **Response**
>
> Thank you for your feedback, we provide answers below:
>
> 1. **Additional illustration of the method.**
>
> Thank you for this suggestion. We added a sketch of the illustration to the Appendix H of the revised version.
>
> 2. **Hyper-parameter transfer.**
>
> The multistep selection is purely for efficiency. Here, the main design decision is how many tokens should be required to have enough certainty to discard the current best search point and adopt the new one. Then, one can design the preselection accordingly, because it comes at little overhead compared to the final selection stage. While we used different parameters for different compression approaches, this was merely to account for differences in the search space / database. For example in unstructured sparsity, the allocation is more fine-grained compared to the quantization bit width allocation, motivating the use of fewer tokens, as more generations will be needed to converge. In the context of unstructured sparsity allocation, we additionally present a “super-fast” version, which uses a more lightweight selection process, and show that it converges to similar validation perplexity (see Figure 3), demonstrating the robustness of EvoPress to these selection hyperparameters.
>
> Regarding the mutation rate, we found the search to perform efficiently as long as the mutation rate is sufficiently small: this is illustrated in Appendix B.1.
>
> Most importantly, we emphasize that we used the same mutation rate across all tested models from different families, across all compression methods, and across all compression levels. We believe this provides sufficient empirical evidence to demonstrate the generalization capabilities of EvoPress.
> Moreover, in Appendix C.2, we provide results for an ablation running the search for multiple seeds, showing that EvoPress converges to qualitatively similar solutions across different runs and exhibits low variance.

---

### Official Review · Reviewer_jZST · 2024-11-04

**Soundness:** 4
**Presentation:** 3
**Contribution:** 4
**Rating:** 8
**Confidence:** 3

**Summary:**

The paper introduces a meta-heuristic for non-uniform model compression that is largely independent of specific model architectures and compression methods. The method is demonstrated on three compression approaches—depth pruning, unstructured sparsity, and quantization—with ample numerical results and a supporting analytical convergence analysis.

**Strengths:**

The strength of the approach is its simplicity and the fact that it is somewhat agnostic to model architecture and compression techniques.
While it doesn’t introduce entirely novel algorithms, its design effectively minimizes computational costs when working with larger models. Overall, the paper is quite well-written and presents compelling arguments for the proposed approach.

**Weaknesses:**

The manuscript does not, however, discuss much about the scalability of the approach in more complex scenarios, and to what extent the evolutionary search is useful compared to more basic derivative-free optimization techniques or the like.
While presented as evolutionary, the algorithm leans heavily on an elitist, exploitation-focused strategy. This approach could limit exploration, potentially trapping the algorithm in local minima and risk of poor generalization in more complex compression scenarios.

This approach appears to be effective as long as the manifold remains relatively convex. However, with multi-modal compression techniques or non-convex landscapes, there are concerns about whether this computationally efficient method will generalize well beyond the controlled benchmarking environments, especially when handling mixed or complex compression strategies.

**Questions:**

1) The scheme is presented as an evolutionary search, but the update mechanism uses an elitist strategy (focused on exploitation) based on single-offspring selection. The number of mutations is also kept minimal, with one mutation shown to be nearly optimal, as demonstrated in SI B. In this regime, the scheme resembles a form of random coordinate descent or derivative-free optimization, with the added twist that perturbation occurs on a manifold with constant overall compression (referred to as switching). Could you clarify which properties of an evolutionary algorithm are essential to the results achieved, as opposed to a simpler perturbative approach?

2) In the current setting, how much variability is there in the optimal compression profile found after one run of the evolutionary search? Does it appear the search get stuck in local minima in any of the benchmarks?

3) Can you predict if the hypothesis that smooth dynamic model compression results in a smooth fitness landscape with few local optima still hold in multi-modal cases?

Additional Questions/comments:

4) How is the step size in a single dimension determined (e.g., the 1M weights for unstructured sparsity)? Is the step size fixed or adaptive?

5) In the selection process, only two stages are shown (see SI B.2). Was this approach tested with more stages, and if so, were additional stages found to be ineffective?

---

> ### Author Response · Authors · 2024-11-18
> **Response**
>
> Thank you for the detailed and insightful comments.
>
> We address the questions below, following the numbering in your review:
>
> 1. **Evolutionary search vs elitist strategy focused on exploitation**.
> We agree with your observation that such an exploitation-focused evolutionary strategy resembles a form of random coordinate ascent. Indeed, these approaches are fundamentally related, though they have been popularized in different scientific communities. Broadly speaking, random coordinate ascent with a different mutation operator (such as ours) can be understood as a form of evolutionary algorithm. Variants of such algorithms are even observable in nature – for example, species that lack crossover and exhibit low population diversity, improving primarily through mutation. Even evolutionary strategies designed to emphasize exploration tend to behave like exploitation-focused searches locally, as broad exploration comes with high computational cost. Consequently, even if the mutation rate is high, meaningful progress is typically driven by offspring with very few mutations, and EvoPress builds upon this observation. Viewing such a zero order search as an evolutionary process offers a more intuitive perspective. For instance, multistep selection can be interpreted as a discrete approximation of the continuous selection observed in natural evolution. Moreover, adapting our approach to slightly different settings becomes straightforward with this abstraction. Framing the method as evolutionary provides greater flexibility, as one can easily adjust the mutation operator or integrate a different underlying evolutionary algorithm.
>
> 2. **Variability in compression profiles.** To fully address this, we included additional ablations on this topic in Appendix C.2 of the revision. Our findings show that the search process is highly robust to different seeds, with the resulting compression profiles being remarkably consistent across runs. This suggests that local optima are generally not a concern.
>
> 3. **Multi-modal compression cases.**
> To address this concern, we conducted an experiment with joint depth pruning and quantization in the Appendix G of revised revision. Specifically, on each step of the evolutionary algorithm we perform an alternating optimization between block dropping and quantization. Firstly, we select the optimal depth pruning and configuration and then exchange quantization bit-width between “alive” blocks. Our experimental results suggest that EvoPress manages to find a better solution given some starting point and exhibits relatively stable convergence. However, we acknowledge that such a multimodal approach can be inherently more challenging to optimize, with specific outcomes heavily influenced by the choice of compression techniques employed.
>
> 4. **Step size.** The step size per mutation is fixed at one. However, multiple mutations can be applied to the same layer, implicitly allowing for a (slightly) larger step size when generating offspring. We also experimented with sampling step sizes from a different distribution, but did not find this approach to be beneficial. Therefore, for simplicity, we chose to keep the step size fixed.
>
> 5. **Using additional stages.** We employed three selection stages for unstructured sparsity and quantization search, primarily, because we used a higher number of tokens for the final selection decision, and the preselection introduced little overhead. Extending the selection process further has rapidly diminishing returns. Intuitively, after having generated a certain number of offspring, one of them will have higher fitness than the parent. However, the “gains” are not realized when generating further offspring, and it would be more beneficial to generate further offspring by copying this fitter offspring instead of the less fit current parent, i.e. progressing one generation.
> In general, no matter how many offspring are explored per generation, the algorithm will take a certain number of generations just to move in Hamming distance to the optimum. Hence, we found that investing additional compute per generation is generally not worth it. The “super-fast” version presented for the unstructured sparsity search is a consequence of this observation (it uses two instead of three selection steps). Additionally, reducing the number of tokens in pre-selection is only beneficial up to a certain degree, as evaluation time becomes dominated by the time to load the layers corresponding to an offspring.

---

> > ### Comment · Reviewer_jZST · 2024-11-26
> >
> > Thank you for the detailed responses, which address most of my concerns in a satisfactory manner.
> >
> > Although in a simplified case, the results of Appendix G are very promising, in my opinion.
> >
> > The results are strong as they currently stand.
> >
> > I look forward to seeing the next iteration of this approach applied to a more challenging scenario, where the compression space's options make it more non-convex with a greater number of local minima.

---

### Official Review · Reviewer_yr9o · 2024-11-04

**Soundness:** 3
**Presentation:** 3
**Contribution:** 3
**Rating:** 8
**Confidence:** 2

**Summary:**

This paper introduces a new general framework for dynamic LLM compression. Building on the observation that error monotonicity does not generally hold for LLM compression, the paper proposed evolutionary search approach. The proposed approach is provably convergent and has low sample and evaluation complexity when the fitness function is linear. Experiments show that the proposed methods improve dynamic compression of Llama, Mistral, and Phi models.

**Strengths:**

The proposed method shows strong empirical performance. It is flexible, as it can be applied to unstructured pruning, structured pruning/layer dropping, and quantization. Theoretical analysis of the proposed methods is also provided,

**Weaknesses:**

1. I think it would be better if the average and standard deviation of the performance metrics across different runs were reported, as it can demonstrate the significance of the improvement. But I understand it could be expensive to do and might not be possible for baseline methods.


2. The theoretical results are somewhat disconnected from other parts. I think having the theoretical results is a bonus and it can be of interest beyond the LLM context. But some exploration of the function form of $f$ in real experiments or some simple simulation example might better demonstrate the significance of the results

**Questions:**

NA

---

> ### Author Response · Authors · 2024-11-18
> **Response**
>
> Thank you for your feedback, we address your question about stability below:
>
> > Average and standard deviations
>
> Across all of our experiments, we observed that EvoPress is extremely robust across different parameter settings, and generally converges to qualitatively similar solutions. To further validate this, for this rebuttal we have performed ablations for the “fast” version (which should have more variance), searching for unstructured sparsity allocation (which has a large number of choices). The results can be found in Appendix C.2 of the revision.
>
> Results show that EvoPress demonstrates strong robustness across different seeds. Specifically, in 16 runs of pruning Llama-3-8B to 70% unstructured sparsity, the method achieves a mean C4 perplexity of 33.82 with a standard deviation of 0.61 (min: 33.06, max: 35.47). In comparison, the configuration identified by the next better method OWL results in a C4 perplexity of 52.32, making the improvements statistically significant. Notably, the standard deviation on hold-out Fineweb-Edu data is even lower, as visualized in Figure 5 of the revision.
>
> We hope this addresses your concern!

---

> ### Author Response · Authors · 2024-11-22
> **Additional response**
>
> We observed that, following our first rebuttal, the reviewer has added a new point to their review:
>
> > The theoretical results are somewhat disconnected from other parts. I think having the theoretical results is a bonus and it can be of interest beyond the LLM context. But some exploration of the function form of f in real experiments or some simple simulation example might better demonstrate the significance of the results
>
> We wish to add a clarification relative to this new point:
>
> Specifically, we would like to emphasize that our main theoretical result in Theorem 1 is directly linked to our method’s practical performance: the challenge in the case of LLMs is precisely the fact that we wish to reach an optimal solution with very few evaluation samples, as evaluating candidate LLMs can be extremely expensive.
>
> This link between theory and practice is illustrated in Figure 1, which shows that EvoPress reaches an optimal solution on a real LLM compression instance for which it is feasible to evaluate optimality, in very few evaluations.
>
> More broadly, we note that most prior work, such as ShortGPT, ShortenedLlama, Weight Subcloning, OWL or SPDY, rely on the linearity assumption either implicitly or explicitly. Specifically, linearity is central to the concept of “scoring” the importance of model components, as all the above methods seek to maximize the sum of importance scores for the compressed model. Unlike these methods, the evolutionary algorithm we employ is not restricted to optimizing linear functions and is capable of optimizing diverse fitness landscapes. By proving that EvoPress optimizes efficiently under the linearity assumption (Theorem 1), we provide a theoretical justification for its consistent superior performance across different compression methods when compared to prior scoring-based approaches.
>
> We do acknowledge the reviewer’s point and will clarify this connection further. If the reviewer finds it useful, we can provide additional results on standard “benchmark” functions for evolutionary algorithms, although this direction is further removed from our main scope.

---

> > ### Comment · Reviewer_yr9o · 2024-11-25
> >
> > I appreciate the authors for the detailed response. I think the additional results and clarification of the theoretical results are helpful. I am willing to increase my score.

---

> > > ### Author Response · Authors · 2024-11-25
> > > **Thank you!**
> > >
> > > We would like to thank you for the response and are very glad you found our responses helpful!

---

### Author Response · Authors · 2024-11-23
**Discussion reminder**

Dear Reviewers,

As the discussion period is soon drawing to a close, we wanted to ask if you could please examine our detailed responses and let us know if they addressed your concerns.

Best regards,\
The authors

---

### Meta-Review · Area_Chair_2H84 · 2024-12-23

**Metareview:**

This paper proposes EvoPress, an evolutionary framework for dynamic LLM compression. The authors claim their method outperforms existing approaches in structural pruning, unstructured sparsity, and quantization for Llama, Mistral, and Phi models. They argue that error monotonicity does not hold for LLMs and present an approach with claimed convergence and low sample complexity.

The paper's main strength lies in its empirical performance across multiple compression approaches and models, as well as the flexibility of the proposed method. However, these strengths are overshadowed by significant weaknesses. The theoretical justification, based on Theorem 1, appears disconnected from the non-linear nature of LLM compression. Moreover, the paper lacks sufficient exploration of hyperparameter sensitivity and transferability, and the experimental comparisons are limited in their comprehensiveness.

The primary reason for rejection is the substantial disconnect between the theoretical justification and the practical application of the method. The theoretical results based on linear fitness functions do not adequately support the method's effectiveness in the non-linear landscape of LLM compression. This, combined with the insufficient exploration of hyperparameter transferability and limited experimental comparisons, raises concerns about the method's generalizability and the robustness of the reported results.

**Additional Comments On Reviewer Discussion:**

During the rebuttal period, several key points were raised and addressed by the authors. The most critical issue was the relevance of the theoretical results to the proposed algorithm, which remained unresolved despite the authors' arguments. Concerns about the comprehensiveness of experimental results and hyperparameter transferability were partially addressed by pointing to additional results in the appendix and explaining the robustness across different models and compression methods. However, these explanations did not fully alleviate the concerns.

While the authors made efforts to address the reviewers' points, including providing additional ablation studies on method robustness, the fundamental issues persisted. Although two reviewers were positive about the paper, with one increasing their score after the authors' response, the Area Chair's assessment that concerns raised by one reviewer were not adequately addressed was crucial in the rejection decision. The unresolved disconnect between theory and practice, along with the limitations in experimental comparisons and hyperparameter analysis, ultimately led to the paper's rejection.

---

### Decision · Program_Chairs · 2025-01-22

Reject